# Carbon–Water Flux Coupling Under Progressive Drought

Sven Boese[1], Martin Jung[1], Nuno Carvalhais[1,2], Adriaan J. Teuling[3], and Markus Reichstein[1]

[1]Max Planck Institute for Biogeochemistry, Hans-Knoell-Strasse 10, 07745 Jena, Germany
[2]CENSE , Departamento de Ciências e Engenharia do Ambiente, Faculdade de Ciências e Tecnologia , Universidade NOVA de Lisboa , Campus de Caparica , 2829-516 Caparica, Portugal
[3]Wageningen University & Research, Hydrology and Quantitative Water Management, Droevendaalsesteeg 3, 6708 PB Wageningen, Netherlands

**Correspondence:** sboese@bgc-jena.mpg.de

**Abstract.**

Water-use efficiency (WUE), defined as the ratio of carbon assimilation over evapotranspiration (ET), is a key metric to assess ecosystem functioning in response to environmental conditions. It remains unclear which factors control this ratio during periods of extended water-limitation. Here, we used dry-down events occurring at eddy-covariance flux tower sites in the FLUXNET database as natural experiments to assess if and how decreasing soil-water availability modifies WUE on ecosystem scale. WUE models were evaluated by their performance to predict ET from both the gross-primary productivity (GPP), which characterizes carbon assimilation at ecosystem scale, and environmental variables. We first compared two water-use efficiency models: The first was based on the concept of a constant underlying water-use efficiency; The second augmented the first with a previously detected direct influence of radiation on transpiration. Both models predicting ET strictly from atmospheric covariates failed to reproduce observed ET dynamics for these periods, as they did not explicitly account for the effect of soil-water limitation. We demonstrate that an ET-attenuating soil-water availability factor in junction with the additional radiation term was necessary to accurately predict ET flux magnitudes and dry-down lengths of these water-limited periods. In an analysis of the attenuation of ET for the 31 included FLUXNET sites, up to 50% of the observed decline in ET was due to the soil-water availability effect we identified in this study. We conclude by noting the rates of ET decline differ significantly between sites with different vegetation and climate types and discuss the dependency of this rate on the variability of seasonal dryness.

## 1 Introduction

The interaction of the global carbon and water cycle has emerged as a critical topic in Earth system science (Ito and Inatomi, 2012; Hartmann et al., 2013). In terrestrial ecosystems, transpiration and photosynthesis are closely linked by gas diffusion in plant stomata (Cowan and Farquhar, 1977; Ball et al., 1987), while the lack of water is one of the principal limiting factors for the productivity of terrestrial ecosystems. Ecosystems can experience perpetual water-limitation, seasonal water stress or irregularly occurring droughts. Climate change is expected to exacerbate existing water scarcities, with a particular increase of drought events expected in Mediterranean regions (Hoerling et al., 2012; Sheffield et al., 2012). Drought events are important

for biogeochemistry because they have been identified as primary sources for the variability of carbon and water fluxes at ecosystem-level (Zscheischler et al., 2014). This can mainly be attributed to a decline of the gross primary productivity (GPP) under severe water limitation (Ma et al., 2012; Stocker et al., 2018) due to limited photosynthesis (Quick et al., 1992; Ort et al., 1994). Despite the importance, predictions of ecosystem responses to intermittent and severe decreases of water availability

remain tenuous as multiple, interacting processes are involved (van der Molen et al., 2011). Furthermore, systematic studies on drought events are hampered by the limited frequency with which they occur at any given location.

The water-use efficiency (WUE) of plants is a central metric for understanding the mechanisms and trade-offs involved during periods of water limitation. It is defined as the ratio of carbon assimilation and water loss through transpiration, therefore reflecting how liberal or sparing plants expend their bounded water resources. From a physiological perspective, limited water

availability poses a dilemma for plants. If they maintained stomatal conductance, they would risk cavitation, effectively halting the translocation of sugars and nutrients (Manzoni et al., 2013; Sperry and Love, 2015). They therefore have to close stomates before embolism can occur (Martin-StPaul et al., 2017), accepting restricted carbon assimilation (Schulze, 1986) and elevated leaf-temperatures, which has the potential to further limit photosynthesis for certain species (Salvucci and Crafts-Brandner, 2004). This response is triggered also by the soil- and leaf-water potential, mediated by the formation of abscisic acid (Davies

and Zhang, 1991) and results in a relative decrease of transpiration and an increase in water-use efficiency (Schroeder et al., 2001; Anderegg et al., 2017). Intercomparison studies show that global biosphere models try to capture this effect with different model formulations, as the exact magnitudes and interactions of relevant processes remain uncertain (De Kauwe et al., 2013; Verhoef and Egea, 2014).

At the leaf-scale, empirical and optimality-based models can accurately predict stomatal conductance and WUE under well-

watered conditions (Leuning, 1995; Katul et al., 2010; Medlyn et al., 2011). For whole ecosystems and based on flux tower data, research has focussed on how water-use efficiency varies with atmospheric water vapor deficit (VPD), assuming well-watered conditions (Zhou et al., 2014, 2015). Embedded in this is the premise that the *underlying water-use efficiency* (uWUE) of an ecosystem is constant in time. Ecosystem-level analyses of the effect of soil-water limitation on stomatal conductance and WUE are further complicated by the fact that atmospheric and soil droughts typically co-occur, making a separation of

the effects of low VPD and low soil-water availability difficult (Knauer et al., 2015). A preceding study further demonstrated that an additional, independent radiation term improves predicting transpiration from GPP and VPD at ecosystem-level (Boese et al., 2017). In this case, a transpiration component not associated to GPP and VPD could be identified, suggesting that radiation directly controls a share of equilibrium-transpiration (Jarvis and Mcnaughton, 1986). Yet the semi-empirical water-use efficiency models suggested by Zhou et al. (2015) and Boese et al. (2017) may not perform well during droughts, where

water limitation is expected to alter ecosystem functioning qualitatively (Farooq et al., 2009). To assess this, dry-down events can be used as natural experiments during which the ecosystem experiences progressive soil-water depletion and thus stress. Dry-down events are periods of many consecutive dry days during which ET declines approximately exponentially with time reflecting an approximate linear relationship between the rate of ET and the remaining plant-available soil-moisture at each time step (Williams and Albertson, 2004; Teuling et al., 2006).

In this study we use a large global archive of flux tower observations containing 31 sites with 47 dry-down events to evaluate WUE formulations during periods of increasing water limitation. We scrutinized how the formulations capture changes in WUE by modelling the observed dynamics of ET. By contrast, using WUE as target variable has the limitation that periods with very high WUE can be the result of both very low GPP and ET, as long as the former exceeds the latter in relative terms. By predicting ET with GPP as an indepenent variable, we avoid overweighting high WUE during very low ET periods, when observational noise could further amplify uncertainties in posterior parameters. While this formulation conceives of ET as an effective output driven by environmental and biological factors such as GPP and VPD, a reverse formulation predicting GPP would also be possible. In both cases, a possible decoupling of GPP and ET (Drake et al., 2018) will then result in lower model performance and can consequently be interpreted as a deficiency of the respective model.

In our analysis, we pay particular attention to systematic biases of ET predictions that impact the predicted dry-down speed. To this end, we show how a simple parameterization based on an effective water-balance-based variable helps in improving predictions under progressive drought. Finally, we assess how the rates of declining ET during dry-down events differ between vegetation and climate types.

## 2 Methods

### 2.1 Data & preprocessing

Observation-based products of gross primary productivity (GPP) and evapotranspiration (ET) obtained with the eddy-covariance method were taken from the La Thuile (open and fair use data policy sites) and Berkeley (Tier 1 data policy sites) collections of the FLUXNET (Baldocchi et al., 2001). Further, we used the global radiation ($Rg$), vapor pressure deficit (VPD) and precipitation ($P$) measured at the corresponding eddy-covariance (EC) sites. Day-time values of GPP, ET and VPD were derived by aggregating observations with potential radiation larger than 10 W m$^{-2}$.

The EC data were pre-processed with established quality-assurance and quality-control (QA/QC) procedures to ensure consistent quality of the observations (Papale et al., 2006). Eddy-covariance GPP values were obtained with the flux partitioning method of Reichstein et al. (2005). We omitted periods not gap-filled with high-confidence, which is particularly important for periods such as dry-down events, as they may represent significant deviations from regular ecosystem behavior. For our analyses, we included data fulfilling a set of minimum conditions: GPP > 0.1 gC d$^{-1}$ m$^{-2}$, ET > 0.05 mm d$^{-1}$ and VPD > 0.001 kPa. This reduces the proportionally large impact of random measurement errors when the observed fluxes are low. As proposed by Beer et al. (2009), we excluded the data for days with a precipitation event (P > 0.2 mm d$^{-1}$) and the three following days. This can reduce contributions by evaporation to the observed evapotranspiration, because physical evaporation typically decreases rapidly after rain events due to the depletion of water stored on leaves (Miralles et al., 2010) and the topsoil (Wythers et al., 1999). Thus, the observed evapotranspiration after three successively rain-free days can be expected to approximate transpiration.

## 2.2 Detection of Dry-Down Events

The identification and selection of dry-down events required special attention. To obtain data that could be confidently assumed to be affected by soil-water limitation, we employed a selection procedure consisting of the sequential application of multiple conditions:

1. Periods with at least 15 successive days without precipitation.

2. Both evapotranspiration (ET) and the fraction of latent energy over net radiation ($ET/R_n$) had a significant negative trend over the course of the precipitation-free period. The latter ratio ensures that declining availability of energy is not responsible for observed declines in ET.

3. ET had to be be controlled more by the diminishing supply of water rather than atmospheric demand.

The latter condition was implemented by combining two models that individually represented demand and supply limited ET. For the demand limitation, ET was predicted as a linear function of $Rg$

$$\mathrm{ET} = a \cdot Rg + b, \tag{1}$$

where $a$ and $b$ are estimated regression parameters. For the supply limitation, ET was predicted as an exponential decrease with time:

$$\mathrm{ET} = \mathrm{ET}_0 \cdot e^{-k \cdot t}, \tag{2}$$

where $\mathrm{ET}_0$ denotes a parameter for the initial rate of ET at the beginning of the exponential decrease and $k$ denotes the rate of the decay. The variable $t$ denotes the days since the beginning of the selected period.

The demand model was applied to the beginning of any period fulfilling conditions 1. and 2. until a time $t = t_\alpha$, while the supply model was applied to the rest of the period. To find the time step after which supply limitation dominated ET dynamics,
we initially set $t_\alpha = 5$ to allow at least 5 observations to be fitted with the demand model and all subsequent ones with the supply model. The residuals of both models were concatenated and the root mean squared error (RMSE) was calculated. We then increased $t_\alpha$ by daily increments until the period fitted with the supply model contained only 5 observations. For each change of $t_\alpha$ the RMSE was noted.

The beginning of supply limitation could then be defined as the $t_\alpha$ for which the RMSE was smallest. As any further increase
of $t_\alpha$ would result in a higher RMSE, this indicates that the ET following $t_\alpha$ was best approximated with the exponential decay function which in turn represents supply limitation.

Figure 1 exemplarily shows ET and RMSE for a period fulfilling conditions 1. and 2. The RMSE decreased until $t_\alpha = 12$ and increased gradually thereafter. This means that the ET past $t_\alpha = 12$ could be better predicted with the exponential decrease depending on time rather than the atmospheric demand.

To verify that the selected period did indeed show an approximately exponential decay of ET, we further required that ET had to fit an exponential function with $R^2 > 0.6$.

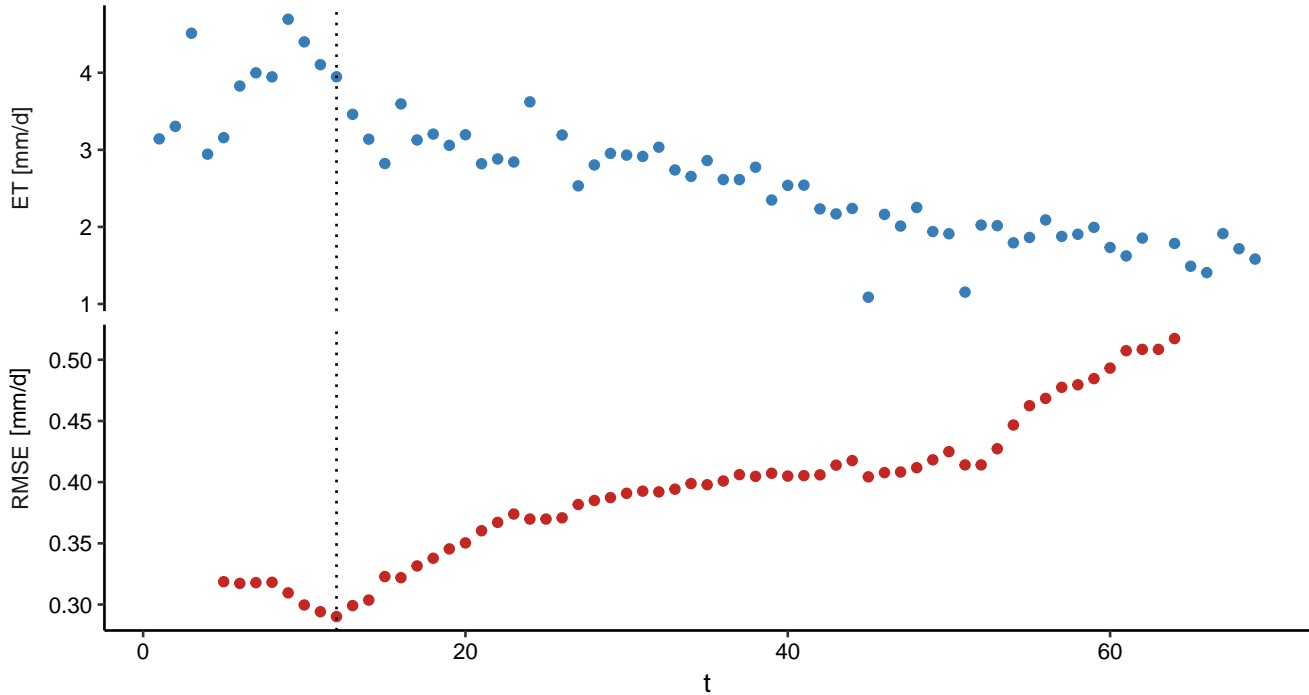

**Figure 1.** Sequences of ET and RMSE used to identify supply limited ET at a dry-down event of the FLUXNET site AU-Dry. The dotted line denotes the day with the smallest RMSE of the combined models, thus indicating the beginning of supply limited ET.

A list of the 47 identified dry-down events detected at the 31 respective sites can be found in the supplementary materials. This table also contains the parameters $a$, $b$ and $k$ used in the detection of the dry-down events (Equations 1 and 2).

### 2.3 Derivation of soil-water availability proxy

Empirical studies that investigate the effects of water availability on ecosystem fluxes across many sites are limited by the
5 availability of consistent estimates of soil-water. To gauge the total amount of plant-available soil-water, measurements would need to sample the complete soil profile in small increments. Even fine-grained measurements cannot remedy a central problem with soil-water observations, namely the quantity does not necessarily reflect the water-stress actually experienced by the plants. This is a particularly severe limitation for studies that aim to associate observed patterns in ecosystem-level fluxes with related changes in the available soil-water. Furthermore, the soil-water contents at specific depths would need to be weighted
10 with the root water uptake which can differ substantially based on root architecture and physiology (Schneider et al., 2010).

The absence of rainfall in conjunction with the observed decrease of ET can offer a valuable opportunity to establish a water-balance based proxy variable in analogy to the "relative extractable water" that is frequently used in ecosystem or land surface models. Conceptually, the magnitudes and rates of decline of ET under a high evaporative demand of the atmosphere can be seen as an integrated measure of the decreasing availability of the soil-water. This means that the proxy for soil-water

availability can be inferred without using any sub-surface measurments, while reflecting the overall soil-water status of the ecosystem.

The amount of water stored in the root zone depends on the mass balance of precipitation, surface runoff, evapotranspiration as well as storage changes due to subsurface runoff. When filtering for precipitation-free periods, we can assume that the

amount of stored water depended solely on the output by observed evapotranspiration, because runoff during these periods can be assumed to be very low. During the exponential decays of dry-down events, the evapotranspiration rate of each time step is defined as a direct product of the available soil-water. At the beginning of a given dry-down event, we assumed that the remaining soil-water, $S_{\mathrm{rem}}$, was equal to an integral of the exponential decay of evapotranspiration:

$$S_{\mathrm{rem0}} = \int_{t=0}^{\infty} \mathrm{ET}_t \tag{3}$$

where $\mathrm{ET}_t$ denotes the evapotranspiration predicted by a fitted exponential decay model. For each successive time step, we then subtracted the respective evapotranspiration from the prior $S_{\mathrm{rem}}$:

$$S_{\mathrm{rem}t+1} = S_{\mathrm{rem}t} - \mathrm{ET}_t \tag{4}$$

If the ET observations had missing values, we used the ET predicted by the exponential decay model instead. Finally, we rescaled the $S_{\mathrm{rem}}$ from its value in mm by dividing it by $S_{\mathrm{rem0}}$, yielding a variable bounded by 0 and 1.

The advantage of this water availability measure is that it can be estimated consistently for dry-down events across diverse ecosystems solely from flux tower data, and that it is constrained by the water balance. A main disadvantage is that the measure can only account for soil-water availability during periods with exponentially decreasing ET. Furthermore, we here assume that the influence of groundwater can be neglected if we observe decreasing ET during periods without precipitation.

In the calculation of $S_{\mathrm{rem}}$, we normalize by the maximum calculated value. Thus, at least one value of $S_{\mathrm{rem}}$ for each site

will be 1. It is important to note that this value must thus not signify unstressed conditions. In the absence of knowning the true extent of the total soil-water storage, this limitation has to be accounted for by calibration of site-specific model parameters.

## 2.4   Models

A water-use efficiency (WUE) model can be formulated as:

$$\mathrm{WUE} = \frac{\mathrm{GPP}}{\mathrm{ET}} = f(x_1, \ldots, x_n), \tag{5}$$

where $x_i$ can include different variables affecting WUE, such as the vapor-pressure deficit (VPD). Evaluating different models against the quotient of GPP/ET has the disadvantage that larger WUE values will be given disproportionate weight when fitting the model. However, these values can occur under conditions with very low GPP and ET, thus having little ecological significance.

To properly test the model with flux tower derived GPP and ET, while accounting for flux magnitudes, we first inverted the model to:

$$\mathrm{ET} = \frac{\mathrm{GPP}}{f(x_1, \ldots, x_n)}, \tag{6}$$

For our analysis, we started with the WUE model proposed by Zhou et al. (2015):

$$\mathrm{ET}_t = \frac{\mathrm{GPP}_t \cdot \mathrm{VPD}_t^{0.5}}{\mathrm{uWUE}}, \tag{7}$$

where uWUE denotes the site-specific *underlying water-use efficiency* assumed to be constant in time and can thus be estimated as an empirical parameter using statistical regression techniques. For increased clarity, variables are henceforth labelled with a subscript $t$, indicating that they vary with time. Recently, Boese et al. (2017) found that radiation is an important driver of transpiration, independent of gross primary productivity. While this model was derived primarily as a response to systematic errors of the model by Zhou et al. (2015), a direct response of transpiration to radiation has been posited before (Jarvis and Mcnaughton, 1986). In this concept, one component of transpiration is driven by the gradient of the vapour-pressure deficit (imposed transpiration), while the other is driven by the radiative energy input (equilibrium transpiration). While water-use efficiency models based on stomatal conductance theory typically can account for the former part, they neglect the latter. Therefore, we formulated an amended version of the model by Zhou et al., further referred to as "Rad":

$$\mathrm{ET}_t = \frac{\mathrm{GPP}_t \cdot \mathrm{VPD}_t^{0.5}}{\mathrm{uWUE}} + r \cdot Rg_t, \tag{8}$$

where Rg denotes incoming solar radiation and r denotes a site-specific parameter controlling the radiation-associated equilibrium transpiration.

Both models outlined in Eqs. 7 and 8 do not explicitly account for the limiting effect of soil-water availability on transpiration. Indirectly, however, this effect is partly contained in one of the predictor variables, the GPP: With decreasing soil-water content, plants may contract their stomata to avoid water loss. This would inevitably lead to a reduction of $CO_2$ diffusion into the leaf and subsequently an inhibition of photosynthesis. The GPP does thus contain information regarding the soil-water status during dry-down events. However, predicting ET from GPP assumes that soil-water limitation affects both GPP and ET equally, while studies suggest reductions of xylem conductivity during drought (Ladjal et al., 2005). Such a reduction would however not necessarily affect GPP in equal measure. To model an *explicit* effect of the soil-water availability on transpiration, we used a stress scalar $s$ adopted from Keenan et al. (2010):

$$s = \left( \frac{S_{\mathrm{rem}t}}{\max(S_{\mathrm{rem}t})} \right)^q, \tag{9}$$

where $q$ denotes a site-specific shape parameter that modifies the response of $s$ to $S_{\mathrm{rem}}$. For both the Zhou and the +Rg models the resulting evapotranspiration was then calculated as the product of the unattenuated model predictions with the

attenuating factor $s$ reflecting soil-water limitation (SWL) as

$$\text{ET}_t = s \cdot \left( \frac{\text{GPP}_t \cdot \text{VPD}_t^{0.5}}{\text{uWUE}} \right) \tag{10}$$

for the Zhou+SWL model and as

$$\text{ET}_t = s \cdot \left( \frac{\text{GPP}_t \cdot \text{VPD}_t^{0.5}}{\text{uWUE}} + r \cdot Rg_t \right) \tag{11}$$

5    for the +Rg+SWL model.

## 2.5    Model calibration and evaluation

All models were evaluated against ET observations by comparing measured with predicted values in a cost function. The parameters were estimated with a two-step algorithm to avoid local minima: First a pseudo-random search within defined bounds followed by a Levenberg-Marquardt gradient-based search (Moré, 1978). In both steps, the cost was defined by the 10  sum of squared deviations.

We evaluated the models with multiple different metrics. A variant of the Nash-Sutcliffe model efficiency (MEF) was used as the primary criterion to assess the accuracy of the predictions (Nash and Sutcliffe, 1970). It is defined as:

$$\text{MEF} = 1 - \frac{\sum \left( Y_{\text{prd}} - Y_{\text{obs}} \right)^2}{\sum \left( Y_{\text{obs}} - \overline{Y_{\text{obs}}} \right)^2}, \tag{12}$$

where $Y_{\text{obs}}$ denotes the observations of a variable $Y$ and $Y_{\text{prd}}$ denotes the predictions. The maximum value of 1.0 is reached 15  in case of a perfect agreement between the observations and predictions. This metric is related to the $R^2$, however it has the advantage that the bias of a model is integrated. To avoid that very large negative values have a disproportional impact on averages calculated across sites, we rescaled negative MEF with:

$$\text{MEF}_{\text{bounded}} = \begin{cases} \text{MEF} \geq 0 : \text{MEF} \\ \text{MEF} < 0 : e^{2 \cdot \text{MEF}} - 1 \end{cases} \tag{13}$$

which yields a $\text{MEF}_{\text{bounded}}$ that exponentially approaches $-1$ in the negative infinite limit. In the following, we refer to 20  $\text{MEF}_{\text{bounded}}$ as MEF for simplicity.

To assess differences of metrics between models, calibration schemes or classes of site characteristics, we used bootstrapping to derive 95% confidence intervals for the respective metric (Efron, 1979).

To assess the ability of the models to reproduce the over-all trends during dry-down events, we also calculated coefficients of the exponential decay (Teuling et al., 2006). We assume that a dry-down event follows an approximately exponential behavior 25  of the form

$$\text{ET}_t = \text{ET}_{t=0} \cdot e^{-k \cdot t} \tag{14}$$

The coefficient $k$ denotes the slope of the exponential function. If this form is assumed to be the general form for dry-down events, then $k$ reflects the rate at which ET decreases. A higher value of $k$ would then indicate a faster rate at which ET decreases over time. This parameter can be used as an index for assessing whether water-use efficiency models correctly reproduce the rate at which ET declines during a dry-down event. For many droughts in the FLUXNET database, ET exhibits a distinctly exponential decrease indicating that availability of soil-water becomes the predominant control of the flux (Fig. 2).

## 2.6 Experimental design

The models outlined in Eqs. 7, 8, 10 and 11 were evaluated in two different *evaluation schemes*:

1. In the first, both the Zhou and +Rg model were calibrated to the full record of suitable observations of the site and evaluated for periods without water-limitation, or "unstressed". This evaluation scheme is referred to as **USevl**.

2. The second scheme used the same parameter estimates, however, the models were now evaluated exclusively during dry-down periods. We refer to this evaluation scheme as **DDevl**.

The parameters that were calibrated in the different schemes were uWUE for all models, $r$ for the variants including the additive radiation term and $q$ for the +SWL model variants integrating the water availability proxy $S_{\mathrm{rem}}$. To assess the variability of the predicted lengths of dry-down events between sites, we classified all sites according to their reported biome types into three classes reflecting vegetation height: *Short* included all FLUXNET sites with the biome types GRA (grassland) and CRO (crops). *Tall* included all FLUXNET sites with the vegetation types EBF (evergreen broad-leaf forest), DBF (deciduous broad-leaf forest), ENF (evergreen needle-leaf forest), MF (mixed forest). *Mixed* included all sites with the vegetation types SAV (savanna), WSA (woody savanna), OSH (open shrubland) and CSH (closed shrubland). Due to the preponderance of forest ecosystems vegetation in the *tall* class, our distinction can elucidate the different ecosystems differ in their water-use strategies as a result of with shallower and deeper root networks (Jackson et al., 1996) and the risk of xylem embolisms (Ryan and Yoder, 1997; Koch et al., 2004). Despite the association that such a distinction allows, it has to be considered as an inexact proxy for the height distribution of plants in any given ecosystem. As that variable is not reported consistently across FLUXNET sites, the separation of ecosystems into the listed categories serves as a first, qualitative approximation and has to be interpreted with caution.

Furthermore, we also explored whether the lengths of included dry-down events depended on hydro-climatic properties of the sites:

Firstly, we used the documented Koeppen-Geiger climate classes for the different sites. Due to the limited sample size of this study, we aggregated the climate classes into four categories: *Temperate/continental humid* conatained sites with Koeppen-Geiger classes Cfa, Cfb or Dfa. *Mediterranean* contained those with the classes Csa or Csb. *Semi-arid / Arid* contained those with classes BSk or BSh, while *Savanna* contained sites with class Aw.

Secondly, we used a *water-availability index* (WAI), which is a metric derived as a simple water-balance model with one storage component (Teuling et al., 2006) driven by daily precipitation and potential evapotranspiration obtained from CRUNCEP reanalyses (Tramontana et al., 2016).

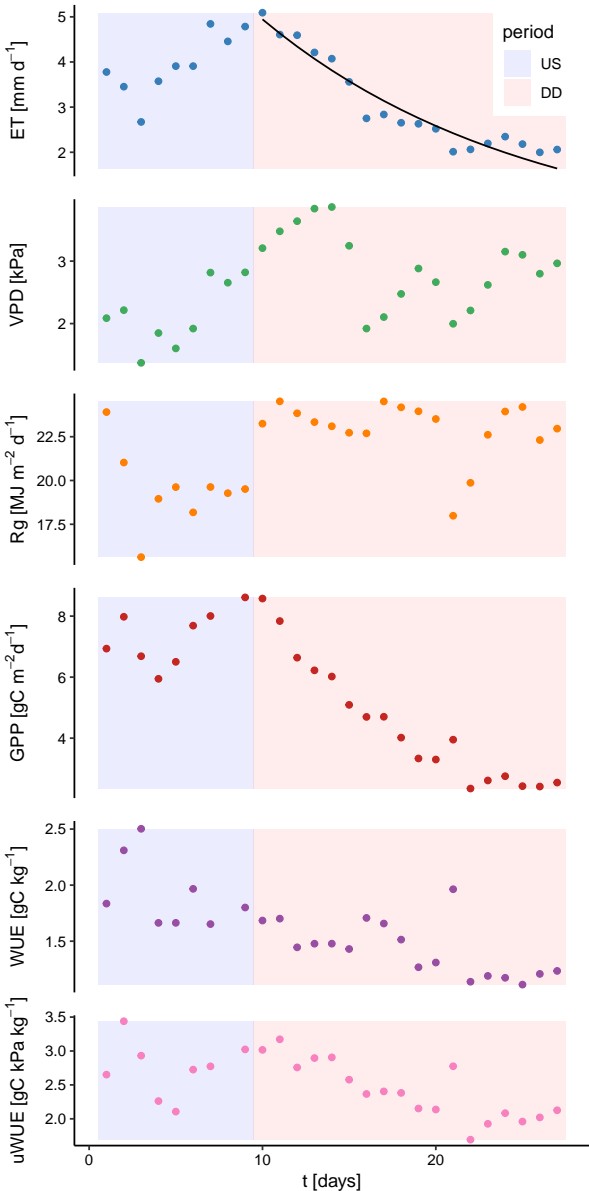

**Figure 2.** Temporal behavior of relevant ecosystem variables during an *unstressed* period (US) and a *dry-down* event (DD) at the FLUXNET site US-Arc. While ET and GPP show a distinct and exponential decay during the dry-down event, the variables reflecting the atmospheric demand (solar radiation, Rg and the vapor pressure deficit, VPD) remain high. The black line denotes an exponential fit to the decreasing ET.

First, each site was initialized with $\mathrm{WAI} = 100\,\mathrm{mm}$ of plant-available soil-water. For each subsequent time step, the output of plant-available water from the ecosystem ($w_{\mathrm{out}\,t}$) was calculated as:

$$w_{\mathrm{out}\,t} = \min\left(\mathrm{PET}_t, k \cdot \mathrm{WAI}_{t-1}\right),\tag{15}$$

where PET denotes the potential evapotranspiration, $k$ denotes the maximum fraction of soil-water available for evapotranspiration without limitation of atmospheric demand. For this calculation, we set $k = 0.05$, which was found as a median value for different ecosystem types by Teuling et al. (2006). The water-availability index for each time step $(\text{WAI}_t)$ was then calculated as:

$$\text{WAI}_t = min(100, \text{WAI}_{t-1} - w_{\text{out}\,t} + P_t) , \tag{16}$$

where $P_t$ denotes the amount of precipitation for each day.

While this index does not incorporate important site-specific characteristics of soil and vegetation, it can serve as climatic measure of water-availability that incorporates basic principles of soil-water dynamics such as seasonal memory-effects and the co-limitation of supply and demand. After deriving mean-seasonal cycles of WAI at each site, we used the interquantile difference $q(0.99) - q(0.2)$ as a measure of the *seasonal dryness* that a site typically experiences for a sufficient fraction of each year.

## 2.7 Fraction of radiation-associated transpiration

The augmented water-use efficiency model described in Eq. 8 can be used to partition the total predicted transpiration into diffusion- and radiation-associated transpiration due to the additive formulation. It is then possible to calculate the fraction of transpiration which was statistically associated with radiation as

$$\text{ET}_{\text{frac}\,t} = \frac{r \cdot Rg_t}{\frac{\text{GPP}_t \cdot \sqrt{\text{VPD}_t}}{\text{uWUE}} + r \cdot Rg_t} \tag{17}$$

where $\text{ET}_{\text{frac}\,t}$ denotes the fraction of radiation-associated transpiration. The parameters $r$ and uWUE are before estimated for the respective site.

## 2.8 Attenuation

Dry-down events were defined and identified by their characteristic decay of evapotranspiration. For many dry-down events, the decline of ET was accompanied with similarly exponential declines of GPP. Due to the strong remaining dependency of ET on GPP, this in itself can explain a certain share of the observed ET decline.

However, in this analysis we posit that an additional attenuating effect may play a role in the temporal dynamic of declining ET. To quantify the magnitude of this effect, we calculate the total fractional reduction of ET that can be attributed to the +SWL term as

$$d = \frac{\sum (1 - s) \left( \frac{\text{GPP} \cdot \text{VPD}^{0.5}}{\text{uWUE}} + r \cdot R_g \right)}{\sum \left( \frac{\text{GPP} \cdot \text{VPD}^{0.5}}{\text{uWUE}} + r \cdot R_g \right)} \tag{18}$$

where the denominator is the summed predicted ET of the +Rad model without limitation factors, while the numerator represents the sum of daily ET reductions as a result of using the vector $s$ in the +Rad+SWL model (Eq. 9).

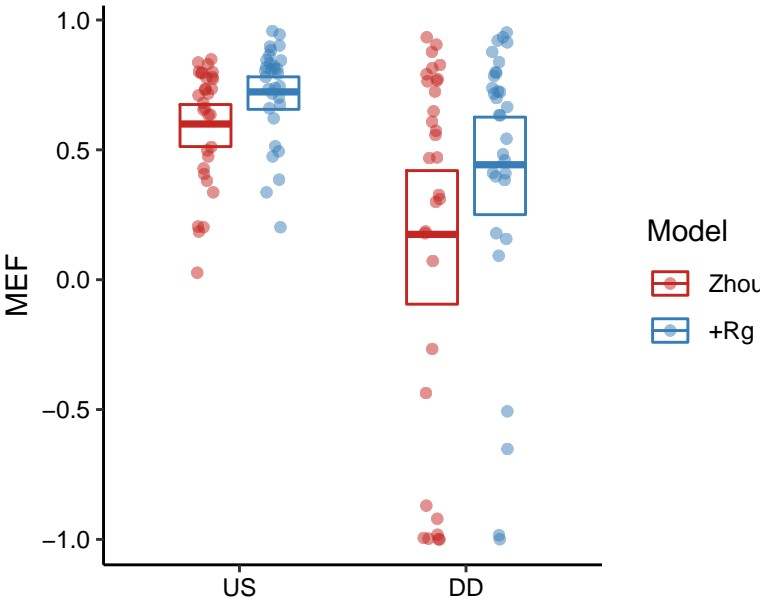

**Figure 3.** Model-efficiency of the Zhou and +Rg models when evaluated inside (DDevl) and outside (USevl, see *Methods / Experimental design*) water-limited dry-down events. The models were calibrated for all adequate site-observations. The points indicate MEF for individual sites. The bold lines denote the mean, while the boxes indicate the bootstrapped 95%-confidence-intervals of the mean.

## 3   Results

As a first test for the validity of water-use efficiency under water-limitation, we evaluated the Zhou and the +Rg model inside and outside dry-down events with a bounded Nash-Sutcliffe Model Efficiency (MEF). The calibration was conducted for each site separately with all available and adequate observations, irrespective of the soil-water status. Both models showed

significantly and strongly reduced MEF when the models were evaluated during the dry-down events rather than periods without water limitation (Fig. 3). During these periods, the +Rg model still outperformed the Zhou model.

To diagnose the origin of the differences in MEF, we assessed the magnitude of the model residuals over the course of DDEs. Aggregated across all dry-down events, model residuals declined systematically with increasing drought (Fig. 4). For the Zhou model, the absolute residuals were biggest at the beginning of the events. Based on the characteristic dynamic of the model

residuals we concluded that merely including the similarly declining GPP as predictor was insufficient to predict ET during these periods. Specifically, both models tended to underestimate ET at the beginning of the dry-down events, when soil-water can be assumed to be in ample supply. Towards the end, when soil-water has become considerably more limited, the +Rg model tended to overestimate ET.

As we noted, during dry-down events, GPP can show exponential declines similar to ET (Fig. 2). This raises the question

why predicting ET using GPP introduces systematic model errors. We thus plotted both the water-use efficiency (WUE) and

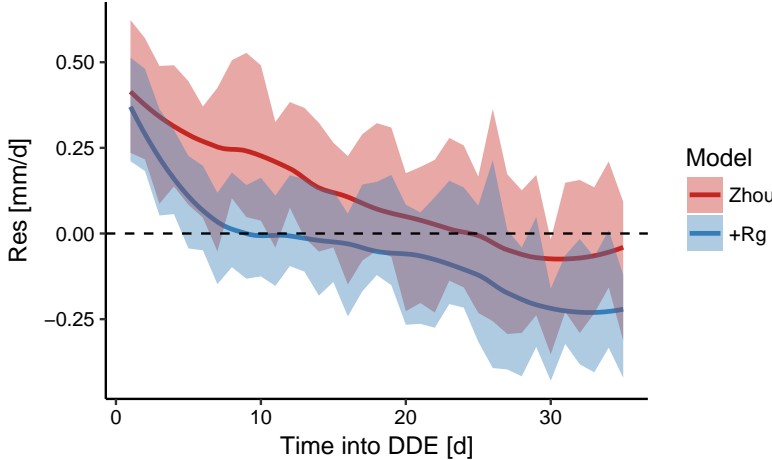

**Figure 4.** Mean model residuals across all dry-down events for the Zhou and +Rg model as a function of time during dry-down events. The shaded area reflects the 95%-confidence-intervals. Both models tended to underestimate ET near the beginning of the dry-down event while overestimating the flux with increasing length of the event. The former was more pronounced for the Zhou model, the latter more pronounced for the +Rg model.

the underlying water-use efficiency (uWUE) against the $S_{\text{rem}}$ variable reflecting the soil-water status of the ecosystem (Fig. 5). As we noted previously, all included models include the response of WUE to VPD and thus do not assume that WUE is constant over time. VPD can be expected to rise during dry-down events, as the moisture supply from the soil and biosphere gradually diminishes. However, we observe an inverse tendency, in which WUE is, on average, higher when $S_{\text{rem}}$ falls below

0.5. Thus, even when accounting for the effect of VPD, uWUE does not remain constant with regard to $S_{\text{rem}}$.

To address the decreased model performance during dry-down events, we provided each model with a mechanism to attenuate transpiration in response to decreases in $S_{\text{rem}}$. Consequently, the original as well as the amended models were reassessed (Fig. 6). The reference Zhou model shows the lowest mean MEF when averaged over sites. Notably, however, for the mean MEF across sites, no significant improvement resulted from adding the effect of soil-water limitation to this model (Zhou+SWL).

The model variants including radiation performed substantially better. For the +Rg variant, including the effect of soil-water limitation paid off with a substantially increased mean MEF. The results indicate that only the combination of radiation and soil-water limitation provided the best predictions of ET during dry-down events.

The coefficient $k$ quantifies the rate of the exponential decay during the dry-downs. Small values indicated a slow decay of evapotranspiration with time. Motivated by the findings of the change in MEF, we contrasted the $k$ values calculated from

the observed ET with those of the ET that the models predicted (Fig. 7). The Zhou yielded more accurate decay rates $k$ when the effect of soil-water limitation was explicitly accounted for. For both variants, $k$ values were unbiased when compared to estimates derived from the observations. By contrast, the +Rg model underestimated $k$ significantly, implying that the predicted

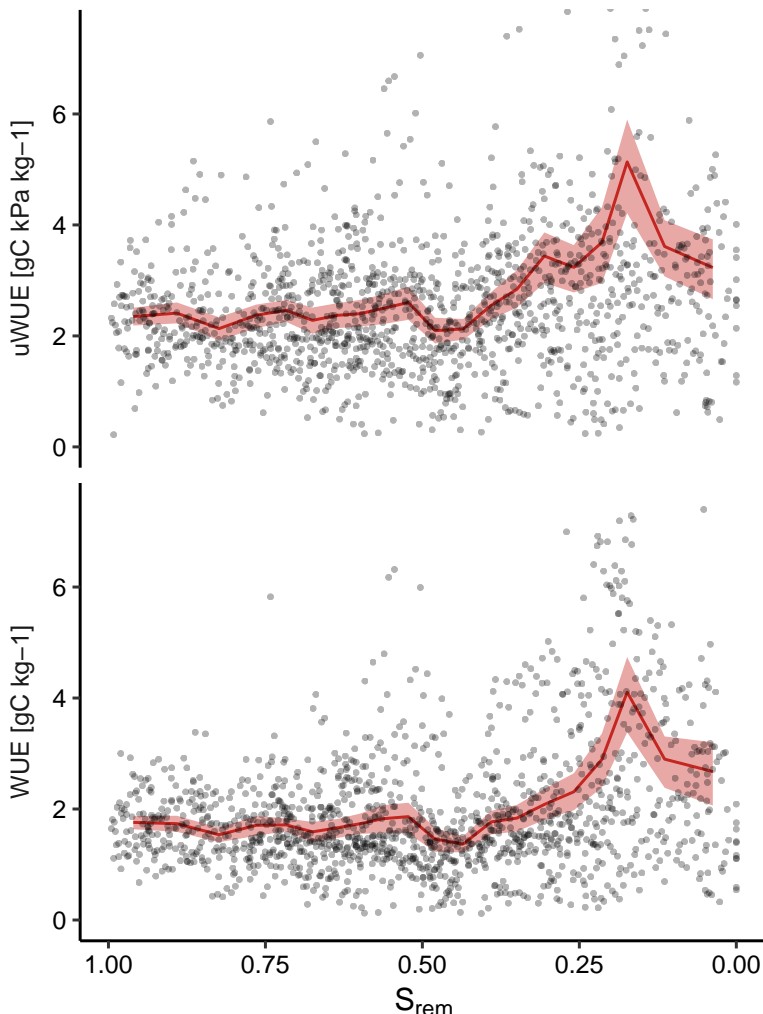

**Figure 5.** Response of the water-use efficiency (WUE) and the underlying water-use efficiency (uWUE) to the remaining soil-water ($S_{rem}$). The points represent daily observations pooled from all sites, while the red line with the shaded area denote the mean response and its bootstrapped 95%-confidence-interval.

ET didn't decline fast enough while the dry-down events were continuing. However, once the effect of soil-water limitation was included in the +Rg+SWL was included, the $k$ estimates were comparatively accurate and unbiased.

We further used the +Rg+SWL model variant to evaluate the relative reduction ($d$) of ET during dry-down events due to the introduced attenuation factor included in this model. This analysis was carried out stratifying the results along the vegetation and climate types (Fig. 8). Sites with tall and mixed vegetation had significantly higher relative attenuation of ET compared to sites with short vegetation. However, for both vegetation and climate types, there was substantial variability between the different sites.

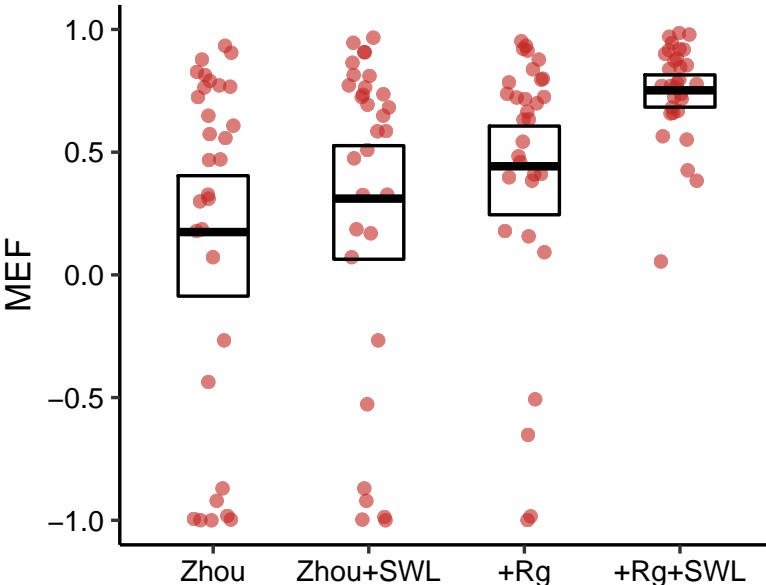

**Figure 6.** Model performance of the two original models and their amended variants which include an attenuation function reflecting soil-water availability. The models were evaluated during dry-down events. The dots denote individual sites. The bold line denotes the mean for all sites, while the box represents the 95% confidence intervals of the mean.

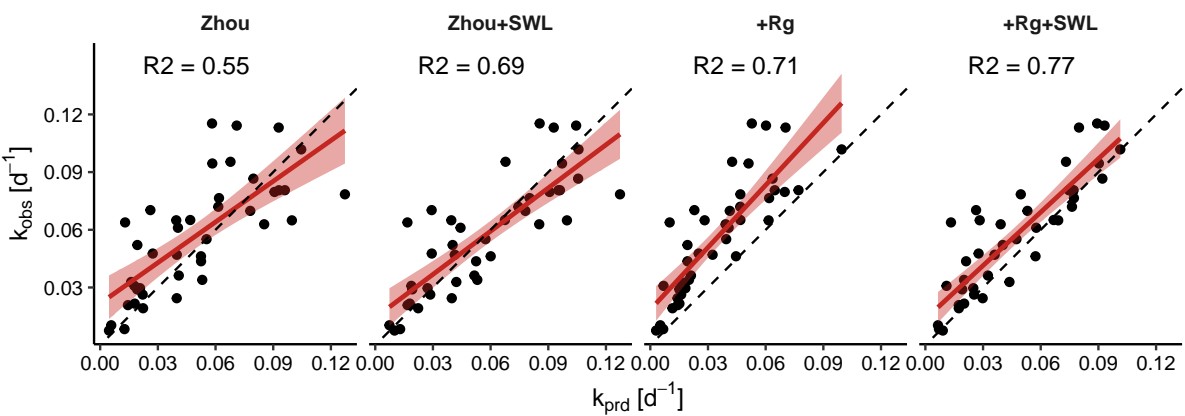

**Figure 7.** Observed plotted against predicted values of the decay-coefficient $k$ (Eq. 14) for the four model variants. Points represent individual dry-down events, for which a linear fit with confidence intervals is shown in red; the one-to-one line is dashed. Three outlying events for which $k$ deviated exceedingly from the other events were removed.

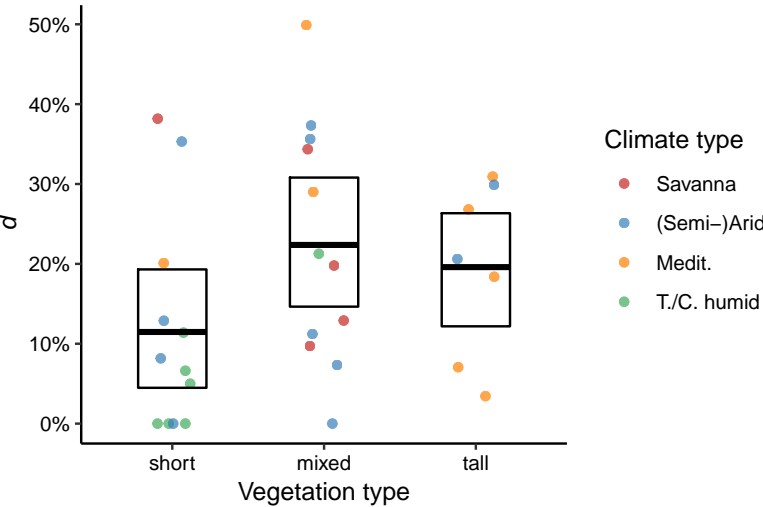

**Figure 8.** Fraction of relative ET reduction during dry-down events ($d$; Eq. 18). The points represent the values for individual sites, the bold bar denotes the mean value of all sites, the box represents the bootstrapped 95%-confidence-interval.

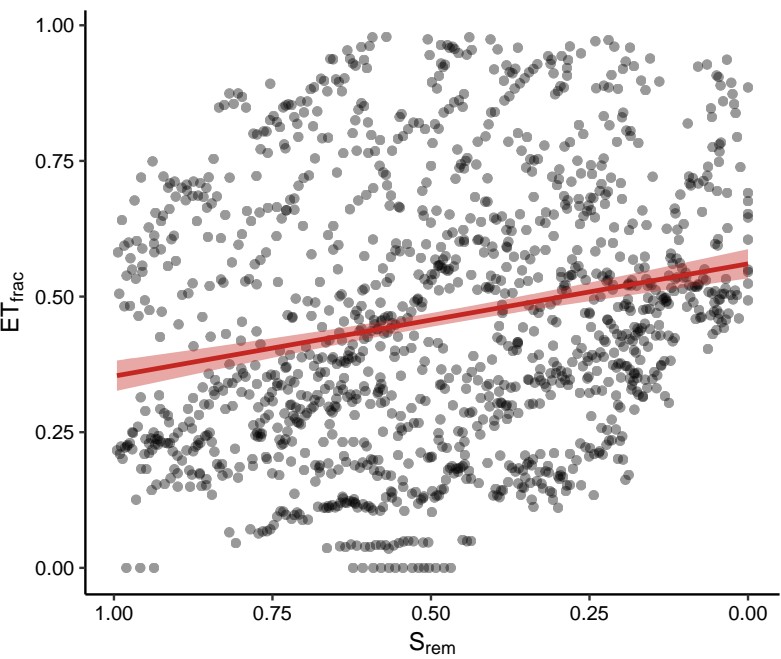

**Figure 9.** Fraction of radiation-associated evapotranspiration ($ET_{frac}$) as a function of $S_{rem}$. Points represent daily values for all dry-down events, the red line is the mean response for all events derived with a linear regression ($p < 0.001$).

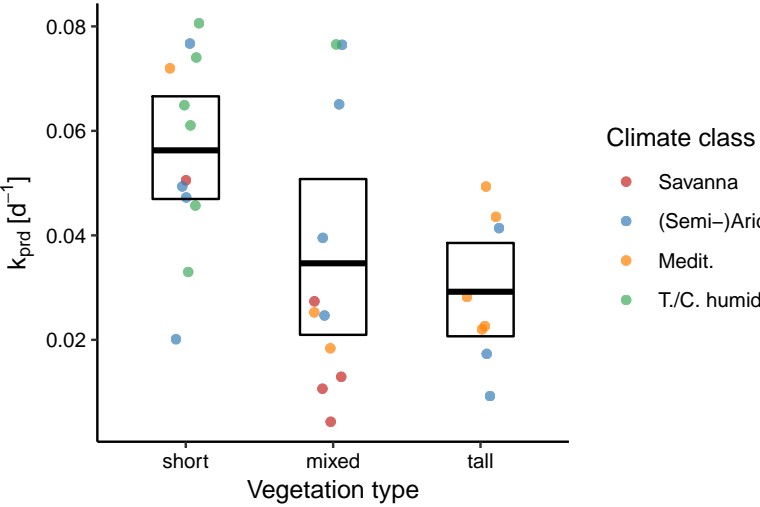

**Figure 10.** Between-site distributions of the predicted decay-rate $k$ stratified along the aggregated vegetation and climate classes. The points represent individual sites, the bold line denotes the mean across sites, the box represents the bootstrapped 95%-confidence-interval.

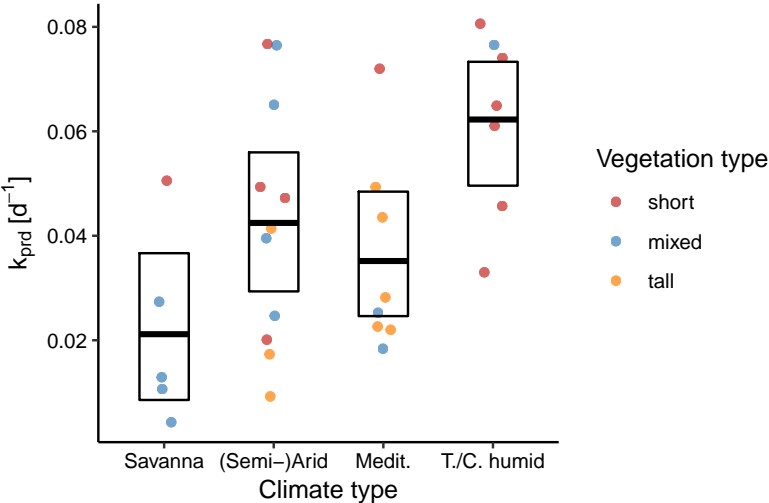

**Figure 11.** Between-site distributions of the predicted decay-rate $k$ stratified along the aggregated climate and vegetation classes. The points represent individual sites, the bold line denotes the mean across sites, the box represents the bootstrapped 95%-confidence-interval.

Boese et al. (2017) proposed a tentative attribution of transpiration to the stomatal conductance and radiation. Here, we analyzed how the fraction of radiation-attributed transpiration ($\mathrm{ET_{frac}}$) changed as a function of the soil-water availability. The daily $\mathrm{ET_{frac}}$ values for all included site varied widely along the observed $S_{\mathrm{rem}}$ (Fig. 9). Despite the substantial variability, the mean $\mathrm{ET_{frac}}$ showed a significant association with $S_{\mathrm{rem}}$; $\mathrm{ET_{frac}}$ was significantly higher for observations with low $S_{\mathrm{rem}}$.

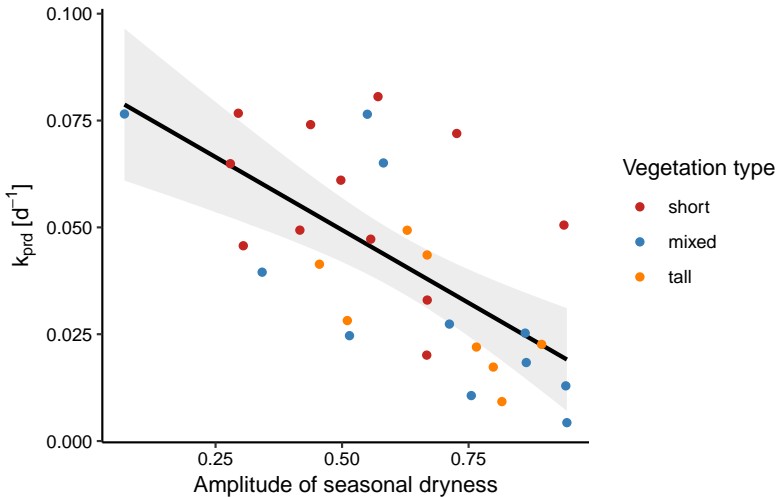

**Figure 12.** The decay-rate $k$ correlated significantly with the mean amplitude of seasonal dryness across sites ($p < 0.001$, $R^2 = 0.42$). Among the vegetation types, this correlation was only significant for the mixed type ($p = 0.007$)

The estimated decay-coefficient $k$ reflects the rate of the exponential ET decline during the dry-down events. We found that $k$ was significantly higher for dry-down events in sites with short vegetation, compared to tall vegetation (Fig. 10), meaning that sites with short vegetation had faster decays of ET during periods of water limitation. However, there was substantial variability within the vegetation types.

5    We also stratified 31 sites along hydro-climatic properties. In a first step, we used an aggregation of Koeppen-Geiger climate classes. We found that *Savanna* climates had the lowest rates of $k$ indicating slowly declining ET. By contrast, sites with a *Continental/temperate humid* climate had the fastest declines, as evidenced by the higher values of $k$. Notably, sites with mediterranean climate tended to have lower $k$, except for one site with low vegetation. As Fig. 11 indicates, the *Semi-arid / Arid* and *Mediterranean* climate classes in particular contain a mixture of plant heights, which complicates inferences regarding

10   the impact of climatic variability on $k$.

Further, we found a significant, negative correlation of $k$ with the amplitude of the seasonal dryness ($p < 0.001$, $R^2 = 0.42$). When separating the three vegetation types, the correlation was signficant for sites with mixed vegetation ($p = 0.007$), yet not those with short or tall vegetation ($p = 0.341$ and $p = 0.801$, respectively; Fig. 12). However, this analysis was severely limited by the sample size within the three vegetation types.

## 4 Discussion

### 4.1 Findings

In this analysis, we showed empirically that water-use efficiency changes during progressive drought are associated with the interaction of radiation and soil-water availability. Merely including the effect of soil-water limitation in a water-use efficiency model without the radiation effect did not improve predictions substantially. By its added effect of soil-water limitation, the +SWL-variant can represent changes in uWUE that occur due to increasing water stress. Our results suggest that such changes were insufficient to lead to significantly improved predictions of transpiration, adding to the finding of a study noting no increase in uWUE for a drought event in an evergreen needle-leaf forest (Gao et al., 2017). In contrast to the model without the radiation term, explicitly including soil-water limitation in the +Rg model led to a significant and substantial improvement of the model performance. This further demonstrates that radiation is required as an important variable for predicting transpiration from GPP and VPD even during water-limited periods, extending the prior analysis that did not explicitly focus on water-limited periods (Boese et al., 2017).

Importantly, established water-use efficiency models assume that the product $GPP \cdot \sqrt{VPD}$ can adequately predict transpiration. Our analysis suggests an ecosystem scale soil-water availability effect on WUE that is statistically independent from VPD effects on the contraction of stomata. With magnitudes of up to 50% of relative ET reduction, its effect was important to predict the rate of ET decline during dry-down events. The presence of the VPD-independent decline underlines the significance of soil-water limitation for ecosystem water-use efficiency during drought. Importantly, the magnitudes of the observed attenuation was siginificantly higher in tall, compared to short vegetation types, indicating that the possibly hydraulic regulation of transpiration during dry-down events is more prominent for these ecosystems. While the reduction of xylem conductivity in drought conditions has been studied (Ladjal et al., 2005) and is a candidate explanation for the observed attenuation, the ecosystem-scale of our analysis does not allow for a definitive association with physiological processes.

Our study posits the countervailing interaction of two additional factors controlling radiation: On the one hand, the positive effect of radiation and on the other hand, the negative effect of soil-water limitation. As we demonstrated in the assessments of model efficiency and the predicted dry-down rates, jointly accounting for both effects was justified on empirical grounds. Despite the effectiveness of the Rad+SWL model at ecosystem-scale, physiological studies under controlled conditions are needed to disentangle the interacting processes.

Further, the rate of the exponential ET decline differed significantly between short and mixed/tall vegetation types, where short vegetation had on average faster declines of ET, consistent with the observations by Teuling et al. (2006). The associated vegetation types, e.g. grasslands and croplands, tend to be dominated by annual plants with shallower root networks (Jackson et al., 1996). These plants could favor fast, relatively unabated transpiration while competing for a quickly diminishing resource. Conversely, tree species dominating the high and present in the mixed vegetation sites have deeper root-networks and would be more circumspect in their water-use to avoid the risk of cavitation which would jeopardize their survival and seed production (McDowell et al., 2008). Similar contrasts of the evapotranspiration response to drought between trees and grasses have been observed for ecosystems were the two plant types co-occur (Baldocchi et al., 2004).

Juxtaposing faster declines and lower attenuation in low vegetation types requires reconciliating both seemingly contradictory observations. In the low vegetation type domininated by grasses, rapidly declining GPP seems to be largely sufficient to predict the also quickly diminishing ET. For the tall vegetation type that is dominated by trees, more gradual, possibly hydraulic limitations could lead to a shallower decline of ET. At the same time, a deeper root zone can sustain ET for comparatively longer periods, thus resulting in lower $k$ values.

Furthermore, we detected a significant correlation between the decay-rate of ET during dry-down events and the mean amplitude of seasonal dryness. Sites experiencing stronger amplitudes of seasonal dryness had lower decay-rates, while the opposite was true for sites with low seasonal dryness variability. Our findings are consistent with the expectation that sites with highly variability in the plant-available water during the growing season have developed adaptations that prevent excessive water stress (Schwinning and Ehleringer, 2001), further replicating Teuling et al. (2006). One likely adaptation in seasonally dry biomes are deeper root networks that allow for sufficient water supply and can potentially tap ground water (Kleidon and Heimann, 1998; Fan et al., 2017). By contrast, ecosystems with low variability of plant-available water have little such adaptations, which are costly from a plant-economical perspective.

The presented results further imply that at ecosystem scale, radiation-associated transpiration (Boese et al., 2017) remains an important process for water-use efficiency models during dry-down events. In fact, we found that the relative share of radiation-associated transpiration increased significantly over the course of dry-down events. Stomatal conductance was responsible for the majority of ET decline during dry-down events, as indicated by a marked decline of GPP during these periods. Toward the later stages of a dry-down event, transpiration was therefore dominated by the part that was not further reducible by stomatal regulation.

## 4.2   Uncertainties & limitations

In this study, we compared the capacity of different semi-empirical water-use efficiency models to predict ET during dry-down events. Previous studies have demonstrated the utility of this approach in identifying patterns and driving factors of ET on different time scales (Zhou et al., 2014, 2015; Boese et al., 2017; Nelson et al., 2018). In these models, the model structure is based on underlying physiological theories and can be amended based on observed model deficiencies. By contrast, the model parameters are calibrated to individual eddy-covariance sites as they are understood to represent constant ecosystem-functional properties (Reichstein et al., 2014). As we outlined in this study, we could evaluate different models and attribute differences in the performance to the inclusion of particular model terms. Because these terms can be linked to physical processes such as equilibrium transpiration (Jarvis and Mcnaughton, 1986) and limitations of hydraulic conductivity (Ladjal et al., 2005), differences between model performances signify the importance of these processes at ecosystem-scale. Nevertheless, the empirical nature and the site-specific calibration of the models can limit which inferences can be drawn from the results. Yet in our comparative approach, some models failed to provide sufficient goodness-of-fit to observed variables even when calibrated, thus allowing a consistent and informative comparison. The calibration to individual sites becomes limiting not for model selection, but rather when calibrated parameters have to be extrapolated while they could be influenced by multiple interacting and incompletely understood processes. In light of both the observed patterns and the limitations of the employed

methodology, experiments under controlled conditions of radiation and soil-water potential could thus shed light on how both variables interact with plant-specific properties to control water-use efficiency under drought.

Despite its demonstrated utility, the new soil-water proxy is also a source of uncertainty in this analysis. The $S_{\text{rem}}$ variable is contingent on the assumption that the decay of ET during dry-down events can be approximated with an exponential function to allow for easy integration. This corresponds to a simple water-balance model with one storage compartment, therefore neglecting both lateral and vertical flow components such as interactions with ground water. Since flux tower observations are largely confined to flat terrain, lateral water fluxes can be possibly neglected here. Potential interactions with ground water may play a role for some sites which would bias the $S_{\text{rem}}$ values low. However, ecosystems in which plants can access groundwater would also less susceptible to declining ET during rain-free periods. Deviations of observed ET decline from a truly exponential decay are likely not critical because it would only affect the normalized $S_{\text{rem}}$ to some extent but not its general temporal dynamics.

One inherent limitation of the $S_{\text{rem}}$ metric is its dependency on periods with exponentially declining ET. Thus, the soil-water status of ecosystem preceding dry-down events cannot be directly accounted for. Indirectly, however, antecedent conditions are reflected in the metric. If the drier conditions preceded an identified dry-down event, the already depleted soil-water content would manifest in lower ET at the beginning of the event itself. This would thus also produce a lower integral in the calculation of $S_{\text{rem}0}$, which quantifies the total remaining soil-water for complete duration of the dry-down event. Nevertheless, despite normalizing the maximum $S_{\text{rem}}$ for each site to 1, the time-step in question could already have depleted soil-water. While the site-specific calibration of the parameter $q$ can compensate for these biases, they complicate interpretations of the parameter across sites. Validating and possibly replacing the proxy variable with a quantity based on direct measurements should thus be a focus of future research.

Due to its character as an effective, ecosystem-scale variable, it integrates various factors affecting the availability of soil-water to plants. This includes biological factors, such as rooting patterns and root-water uptake dynamics, and physical factors, such as soil texture. Comparisons of the dry-down behavior for this variable would therefore need to account for soil properties by using measurements of grain-size distributions if ecological patterns are the focus of the respective analyses. In light of its possible limitations, any future work should first try to link $S_{\text{rem}}$ with directly measured soil-water content. Where representations of soil-water content can be derived from mechanistic land surface models, this could provide an important validation of both the proxy variable itself and possible impacts on the presented findings.

Overall, the results of this study are constrained by the sample size of adequate dry-down events in the FLUXNET data base. Compared to studies that can utilize a large subset of observations, our analyses had to be restricted to events occurring infrequently and only at a small subset of sites in the data base. Despite the comparatively small sample size of dry-down events, the bootstrapped confidence intervals indicate that the patterns were robust for the available sample. Yet when analyzing the variability of $k$ between sites, we noted the considerable variability of values within climate and vegetation types. Superimposing both classifications indicates that variability in one classification can be partially attributed to the other. However, a full intersection of both classifications is currently impossible due to the sample size. Thus, the potential of analyses of the between-site variability of parameters could be extended and be made more robust with more events from a larger set of

ecosystems. An increased availability of eddy-covariance sites would also aid disentangling a variety of confounding factors determining the rates of ET decline across sites. The drought-susceptible *Continental/Temperate humid* grasslands with their fast rates of ET decline and the *(Semi-)Arid* climate type with its large within-class variability of $k$ could particularly benefit from an expansion of eddy-covariance sites.

In our analysis, we examined the different factors controlling transpiration rates during dry-down events. The gross-primary productivity (GPP) was thereby used as a predictor variable. In water-use efficiency models based on physiological theories, GPP contributes information about the degree of stomatal conductance. However, research has indicated that the reduction of GPP during periods of water-limitation cannot be entirely attributed to reduction of stomatal conductance alone, e.g. via reductions of mesophyll conductance (Keenan et al., 2009; Zhou et al., 2013). The model Zhou+SWL predicts ET as a function of

both GPP and soil-water limitation. In our conceptualization, the +SWL term serves as a corrective for non-stomatal limitations of ET. However, it is also possible to conceive of the term as correcting for any difference in how soil-water limitation affects ET differently from GPP. As changes in mesophyll conductance will not affect transpiration rates (Barbour et al., 2010), while reduced xylem conductivity alone will not affect photosynthesis and thus GPP. Thus, a notable reduction in mesophyll conductance would likely manifest by counter-acting the reduced xylem conductivity and lead to a higher correspondance between

GPP and ET than we observed.

To better understand the variability of rates of ET decline $k$ between sites, we stratified our sample of included sites along aggregated vegetation and climate types. In the former case, we distinguished sites with *short*, *mixed* and *tall* vegetation. As the average rooting depth (Jackson et al., 1996) as well as the risk of xylem cavitation (Ryan and Yoder, 1997; Koch et al., 2004) are associated to plant height, we employed this distinction to clarify whether variability of $k$ could be associated to

different plant water-use strategies. It is important to note that this approach is primarily the result of the absence of rooting depth observations for FLUXNET sites. However, any such aggregated classification will be limited in its predictive ability by the considerable variability of confounding factors between classes. The mere fact that both rooting depth and risk of cavitation are correlated to plant height has to caution against prematurely associating the observed variability to any one of these factors. Thus, our study should be seen as grounds for studies in which the effects of rooting depth and plant height can be disentangled

and their respective influence on transpiration rates under drought can be quantified.

### 4.3 Implications & outlook

We could demonstrate that a recursively derived proxy for soil-water limitation could be used to detect and mitigate systematic, structural deficiencies in commonly used semi-empirical water-use efficiency models at ecosystem scale. This variable neither requires soil-water observations that are consistent between multiple sampling locations nor questionable assumptions about

root-water uptake and is derived directly on the ecosystem-scale of interest. This is in contrast to in-situ measurements of soil-moisture which are subject to local heterogeneity and therefore require a potentially problematic upscaling from individual sample locations to the flux footprint of the eddy-covariance tower. More research is required to evaluate the utility of this variable for similar ecosystem-level studies. Its validity could be further tested by contrasting it with temporal profile measurements of soil-moisture, where the individual depths are weighted by the root density or other measures of root water uptake. By

its effective character, this proxy variable could see application for research on other ecological or biogeochemical questions that require measures of soil-water availability which are commensurable across different FLUXNET sites and between events.

The findings of this study indicate that previously developed ecosystem-level water-use efficiency models are biased during water-limitation if they lack the interacting effect of radiation and soil-water limitation. We thus provide evidence that soil-moisture stress has a notable effect on the coupling of carbon and water fluxes. If the aforementioned limitations of $S_{\mathrm{rem}}$ can be overcome, this would have significant consequences for semi-empirical models that link GPP and ET on regional and global scales. Accounting for the observed biases is particularly relevant when these models are used for the partitioning of latent heat fluxes into evaporation and transpiration. Partitioning estimates for these models could be systematically biased if the interacting effects of radiation and water-limitation are neglected. Our findings also suggest that attenuating effects of soil-water availability should be carefully examined in biosphere and land-surface models, because accurate predictions of ET decline during water-limitation are pivotal to understand stress-induced vegetation responses during long droughts. Further research should address whether the observed attenuation effect is of physical or biological nature, which has important implications for understanding plant water-use strategies at ecosystem scale.

*Data availability.*  For this study, we used observations of the FLUXNET initiative from sites with an *open and fair use* (La Thuile 2007 dataset) or *Tier 1* (Berkeley 2015 dataset) data policy. The data sets are available at http://fluxnet.fluxdata.org/data/download-data/

*Competing interests.*  The authors declare no competing interests.

*Acknowledgements.*  This work used eddy covariance data acquired by the FLUXNET community and in particular by the following networks: AmeriFlux (US Department of Energy, Biological and Environmental Research, Terrestrial Carbon Program (DE-FG02-04ER63917 and DE-FG02-04ER63911)), AfriFlux, AsiaFlux, CarboAfrica, CarboEuropeIP, CarboItaly, CarboMont, ChinaFlux, Fluxnet-Canada (supported by CFCAS, NSERC, BIOCAP, Environment Canada, and NRCan), GreenGrass, KoFlux, LBA, NECC, OzFlux, TCOS-Siberia, and USCCC. We acknowledge the financial support for the eddy covariance data harmonization provided by CarboEuropeIP, FAO-GTOS-TCO, iLEAPS, Max Planck Institute for Biogeochemistry, the National Science Foundation, the University of Tuscia, Universite Laval and Environment Canada and the US Department of Energy. Furthermore, we thank the database development and technical support from the Berkeley Water Center, Lawrence Berkeley National Laboratory, Microsoft Research eScience, Oak Ridge National Laboratory, the University of California – Berkeley, and the University of Virginia.

We are grateful to Anke Hildebrandt for discussion and feedback on the paper.

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
