# Peer review of "Carbon-Water Flux Coupling Under Progressive Drought"

_Biogeosciences, 2018_

## Referee Comment (RC1) · De Kauwe (Referee) · 13 Dec 2018

Boese et al. use 47 dry-down events from 31 FLUXNET sites to explore the response of WUE to declining water availability. The authors focus on the response of ET in particular and propose a parameterisation to help improve predictions during drought. I think this is a great experimental approach from which we can learn a lot about how the vegetation responds to declining water availability. However, as currently written this manuscript simply isn't clear enough on a number of small but nonetheless, important points. I suspect many of these could be quickly clarified which I think will help improve the manuscript.

In addition, I did have a few more critical points, specifically:

[Figure]

- With the results, I was hoping to gain a deeper insight into how the land surface responses during drought. In particular, I was hoping to learn about differences in dry-down as a function of vegetation types, hydro-climate, frequency of droughts, etc? This feels like an opportunity missed with these data? I think the authors might consider a more nuanced presentation of their findings, but that is of course up to them. Although I will note they stated: "Furthermore, we also explored whether the lengths of included dry-down events depended on hydro-climatic properties of the sites" and I don't really see where they've done this? - The authors propose the need for two additional corrections, one related to radiation and the other soil water availability. I've commented on this below, it feels unnecessary (mechanistically) and a form of an artificial correction, but I'm happy to be corrected on this and keen to read a more thorough justification. - This leads me to ask about non-stomatal limitations? A number of studies (Keenan et al., 2009; Egea et al., 2011; Flexas et al., 2012; Zhou et al., 2013) have highlighted the need for a non-stomatal correction to GPP (which indirectly affects ET) in order to correctly capture observed responses. This isn't commented on here, but I note that the authors seem to be arguing the opposite, that is, there is a need for a more direct correction on ET but that GPP is fine. This could be a worthwhile discussion point? - What role does LAI, or more specifically, leaf turnover play in the modelling done here? Is it possible that some events see leaf area adjustments which could impact on ET fluxes? Again, I imagine hard to show or not show, but it could be discussed as it may be relevant. - I really think it is important that the authors document all their fitted terms, e.g. the terms in the supplementary, otherwise this study isn't reproducible. - Finally, I'm not clear why the authors only focus on ET? The paper frames the question around WUE and so they should also look at the evolution of GPP during a dry-down, shouldn't they? They could easily argue that GPP isn't directly observed and that is fine, but then I think changing the framing more clearly towards ET only, including removing "carbon" from the title, is warranted.

* Egea, G., Verhoef, A., and Vidale, P. L.: Towards an improved and more flexible representation of water stress in coupled photosynthesis–stomatal conductance models,

Agr. Forest Meteorol., 151, 1370–1384, 2011. * Flexas, J., Barbour, M. M., Brendel, O., Cabrera, H. M., CarriquiìĄ, M., DiìĄaz-Espejo, A., Douthe, C., Dreyer, E., Ferrio, J. P., Gago, J., GalleìĄ, A., GalmeìĄs, J., Kodama, N., Medrano, H., Niinemets, UìĹ., Peguero-Pina, J. J., Pou, A., Ribas-CarboìĄ, M., TomaìĄs, M., Tosens, T., and Warren, C. R.: Mesophyll diffusion conductance to CO2: an unappreciated central player in photosynthesis, Plant Sci., 193, 70–84, 2012. * Keenan, T., GarciìĄa, R., Friend, A. D., Zaehle, S., Gracia, C., and Sabate, S.: Improved understanding of drought controls on sea- sonal variation in Mediterranean forest canopy CO2 and wa- ter fluxes through combined in situ measurements and ecosys- tem modelling, Biogeosciences, 6, 1423–1444, doi:10.5194/bg- 6-1423-2009, 2009. * Zhou, S., Duursma, R. A., Medlyn, B. E., Kelly, J. W., and Prentice, I. C.: How should we model plant responses to drought? An analysis of stomatal and non-stomatal responses to water stress, Agr. Forest Meteorol., 182–183, 204–214, 2013.

Abstract

- As written, I feel like it requires a fair amount of prior insight to follow, e.g. "current semi-empirical water-use efficiency models" - could the authors give an example? "with a previously discovered additive radiation term" - zero context; "20–33% of the observed decline in ET was due to the previously unconsidered" - previously, where? - "in junction" -> "in conjunction"

Introduction

- Pg 1, line 20: it would be nice (but optional) to have a few physiological citations alongside the point about GPP decline with water limitations. - Pg 1, line 22: do you mean "tenuous"? This text doesn't make sense to me, sorry. - Pg 2, line 14: "On leaf-scale" -> "At the leaf-scale"? Also, "can accurately predict" - crucially, under well-watered conditions ... - Pg 2, line 17: "uWUE" hasn't been defined, you need to shift underlying back a bit in the sentence. It is also worth explaining how this differs from WUE described above. - Pg 2, line 18: the text about atmospheric and soil droughts

co-occurring ... It reads as if there is an alternative? Surely, as far as the vegetation is concerned these two will always co-occur? If there is plenty of soil water, then even if there is a precipitation drought, it is not a drought for the vegetation. Am I missing something? I assume the point that is being made here is for the need to seperate out the response to VPD vs the response to soil water. I think this could be more clearly articulated here. - Pg 2, line 20-21: in what context? In a coupled model, a change in radiation due to stomatal closure would be an emergent feedback. I assume this is in terms of an empirical model? I think the authors needs to further explain this point as it isn't self-evident. Similarly -> "Yet these water-use efficiency models" - which WUE models? I've no idea what the authors are referring to here/

Methods

- Suggest renaming "2 Detection of Dry-Down Events & Structure of the Analysis" to methods? - 2.1: it is important to note for the reader that GPP, is flux-derived and not a direct observation -> "Observations of gross primary productivity (GPP)..." - Pg 3, line 20-21: I feel this information should be in the main text and not the supplementary? It seems core for the reader. In the supplementary "I" presumably should become "we". "significant negative trend" - statistically? Can the authors also add the fitted terms, a, b and k to their table 1 in the supplementary. - Pg 3, line 24 onwards: this text isn't clear enough - "namely the quantity does not necessarily reflect the water-stress actually experienced by the plants" - what specifically do the authors mean? Do they mean because these data are usually of limited depth, so do not fully reflect the root-zone? - Pg 4, line 3: to what depth are these "available soil-water" data? My reading of the text is that the authors aren't using any soil water data but instead inferring it but the text is confusing to be honest. It would be worth clarifying. - Pg 4, Eqn 1: What about groundwater? This deserves some mention here, if only to highlight it in the assumptions made. - Pg 4, line 20: Again ... the text about the Boese study and radiation requires further explanation. I suggest it is done once and then it could be referred to as done here. I need to read this paper, but my initial reaction is to query

the statement. Why is radiation an important driver of transpiration, independent of GPP? And why Rg and not net radiation? This feels like a form of double counting here (radiation via PAR is a driver of GPP and Rnet is a driver of ET)? Clarifying this in the text would be worthwhile for the reader. - On a related point - what about evidence of the need for a non-stomatal limitation of photosynthesis during drought? How do the authors suggest this factors into their analysis? - Pg 5, Line 1 onwards: "Both models"?? I assume the authors mean eqn 3 and 4? It isn't clear. I don't follow this text - the soil water availability would also have an effect on ET, the reduction in stomatal conductance due to drought would lead to reduced ET. The text as written makes it appear that this only affects GPP. They then propose an empirical correction on transpiration for declining soil water. I fail to see why this is necessary? The ET quantity reflects the soil water availability? I find this quite worrisome, as above with the Rg, this feels like a double correction that isn't warranted mechanistically. - Pg 5, eqn 5: where is q given by site? It needs to be shown to the reader. - "Short" vs "Tall" feels a pretty vague distinction. I think the tall category would have considerable variability and it would be more interesting to consider the results in the context of the actual heights rather than this arbitrary binary classification. I am aware that it is difficult to obtain these kinds of site characteristics, so the authors do not need to do this; however, I think it would be more interesting if they could. - Pg 8, Eqn 11: how does k vary between sites? - How sensitive are the results from eqn 11 to the assumption of a WAI of 100 mm?

Results

- does figure 2 need to be a figure? It strikes me that it could as easily be a table? It might be then preferable to give an example of a time-series between each model evaluation? - Pg 10, line 10: This point about the ET not declining fast enough, would fit with the narrative I presented earlier of the need for a non-stomatal limitation on GPP, which would also reduce gs and so ET. Of course this wouldn't work for this kind of empirical model. The correction (SWL) could be seen as effectively doing this,
although I don't follow the justification for this approach. - The interpretation of figure 8 seems a bit optimistic and at the very least should be justified ("significant association") with statistics. - Figure 9 ... the difference in k is presented in terms of the "height" of the vegetation, whereas in my eyes it could as easily be interpreted as related to rooting depth and/or leaf area. I'd suggest that height as an explanatory of the difference in dry-down doesn't really have a mechanistic interpretation. At the very least the authors should outline what they think it is a proxy for, or state more clearly how height impacts on the rate of dry-down? Are they hypothesising it is via differences in roughness length?

Discussion

- Pg 15, Line 9: "Established WUE models" - which? Please add some citations, this is very speculative. - Pg 15, Line 10: "Our analysis suggests an ecosystem scale soil-water availability effect on WUE that is statistically independent from VPD effects on the contraction of stomata" - This is a big claim, where is this supported in the data, it would be really helpful to link this to the results. Furthermore, the authors need to unpick this further. If it is independent of the response of gs to VPD, can they discuss the mechanisms they are invoking, presumably via the soil water. Why would it be invariant across ecosystems? This would argue against much of the emerging plant hydraulics literature, surely? Or have I simply misunderstood? I actually see they then link this to a hydraulic limitation related to height - which begs the need to be far more detailed in this analysis. In my eyes it is not sufficient to arbitrarily split the vegetation into small and tall and then to invoke a hydraulic explanation. The tall category could conceivably include a range of heights, do the authors know for certain it is largely made up of very tall trees? I am concerned this is pretty speculative to be honest. - Pg 18, Line 12: Again, *which* "previously developed ecosystem-level water-use efficiency models" - this phrasing or similar is frequently used throughout the manuscript. Please clarify what models and support with citations you are discussing. I'm 99.9% sure you don't mean LSMs but it is very unclear.

[Figure]

Martin De Kauwe

---

## Referee Comment (RC2) · Anonymous Referee #2 · 24 Dec 2018

The authors identified 47 dry-down periods in the Fluxnet database to study how evapotranspiration (ET) is affected by decreasing soil-moisture and if simple calibrated semi-empirical equations based on the concept of Water Use Efficiency (WUE) and an index of water availability (S_rem) are able to explain the reduction of ET with decreasing soil moisture. Results show that water availability exerts an important control on declining ET and this effect is different across Fluxnet sites and for Plant Functional Types characterized by short and tall vegetation or experiencing different seasonality of dry periods. Results indicate that only the combination of a WUE model with radiation and soil-water limitation provided very good predictions of ET during dry-down events (P 10 LL4-5, Figure 4) and remarks that soil water availability has an effect on ET and WUE independent of VPD (P 15. LL 10-11). While the addressed problem,

i.e., ET changes with water availability, is not partially new or the findings particularly surprising, I definitely recognize the value of the quantification and in this regard the presented results are new. Especially, quantifying from observational evidence how ET changes during "clean" dry-down periods, as presented in the article is something important. Furthermore, the study is well written and presented. I have a couple of non-critical comments and then mostly specific comments.

The usefulness of the presented analysis is mostly at the diagnostic level since the presented metric is subjected to a local calibration, depends on the calibration period, and it is not evident how could be used beyond the description presented here. This is partially recognized in the final Section 4.3 (P. 18 LL 7-8) but I have the feeling that some of the statement about the utility of this metric (e.g., P 18 LL 14-17) could be overoptimistic due to the local calibration and strong variability across sites. This may be remarked.

The fact that after two decades of flux-tower data collection only 47 events satisfy the criteria imposed by the authors is a bit discouraging, and does not allow many generalization or comparative analyses, as stated by the authors themselves (P 17 LL 24-29), also due to the huge scatter in the results (Fig. 5 an 7). This scatter is probably due to observational uncertainties but also to behavior of the different ecosystems in response to specific dry-down events and to the definition of S_rem (see below). A few additional words on this problem could be added.

One interesting aspect of the work is to evaluate how much ET decreases because water stress affects GPP and how much is independent of GPP. This is partially shown in Figure 7 but despite the concept of WUE is used as motivation in the introduction of the article and in the Equation (3) to (7), all the figures and results show ET only. There is not a representation of how WUE (e.g., GPP/ET) varies with Srem based on observations. I guess this will provide an additional point of view, which is currently hidden in the analysis. A presentation of changes in WUE would also help to solve the apparently contradictory results according to which the ET attenuation is higher in sites

with taller vegetation, but sites with shorter vegetation have a faster decline of ET (P 16 LL 2-5). I think this result can be only explained if WUE changes in a different way between short and tall vegetation in response to soil moisture decline.

I agree with the authors in not using soil moisture directly (not enough representative) but rather use some water balance proxy of it (P 3, LL 28-30). However, the main issue I have with the definition of Srem is that it cannot keep track of any precedent effect of water availability or soil water stress in the system, or in other words, the starting point of the dry-down is independent of the real initial condition and it just depends on the amount of ET occurring afterwards. This is partially recognized by the authors (P. 4 LL14), but I still would like to see some discussion of the potential implications in the discussion section.

Specific comments

P 1. LL 4. It is not very clear at the abstract level, what is meant with "semi-empirical water use efficiency models".

P 1 LL 18. I am not sure if Dominguez et al 2012 is the best reference here, I would search for articles with either a broader geographical perspective or with more focus on the sub-tropical climate that is mentioned.

P 2. LL 7. As a matter of fact, stomatal closure is occurring always at higher potentials than critical cavitation levels for xylem (Martn-StPaul et al 2017).

P. 2 LL 8. Increased leaf-temperature does not necessarily lead to a decrease in photo-synthesis; it depends on the actual temperature and temperature-sensitivity of a given species.

P. 2. LL 15. Also the classic empirical models, not based on optimality, can reproduce stomatal conductance and WUE responses to VPD (e.g., Ball et al.,1987; Leuning 1995)

P. 2. LL 25. Why are you stating that ET and soil moisture are following a linear

relation? Is this following the exponential decrease of ET with time? Then, very likely, the linearity is with some "proxy" values of soil moisture as S_rem and not with the actual soil moisture.

P. 3. LL 20-21. The way you compute dry-down event and especially the separation between daily ET limited by atmospheric demand and by soil moisture is crucial for the rest of the article as also shown by the sensitivity to the calibration period. Therefore, I would strongly encourage to move the Supp. Material 1 to the main text. It is not too long and it is important to have the full methodological explanation at this stage. I was very confused for many pages on "when" the exponential decrease was assumed to start, if immediately at the beginning of the selected period or after a few days.

P. 6 LL 24. Maybe just an impression but it is not very clear what "all models" refer to, an explicit reference to Eq (3), (4), (6) and (7) would be useful.

P. 8 LL 10-17. For how many steps the WAI_t variable is computed? Since the beginning is from the arbitrary 100 mm in order to extract the mean seasonal cycle of WAI, you need several years.

P. 8. LL 19. Given how WAI is computed, memory effects refer only to seasonal effects, since WAI is averaged.

P. 8. LL 25. I am not fully convinced by this definition; actually also the transpiration associated to GPP is linked to radiation even though indirectly. I would suggest to use a different wording and nomenclature for ETfract_t.

P. 9. LL 21. This is very much expected since they do not have any way of accounting for soil-water limitations.

P 13. LL 11-12. This result is a bit counterintuitive to me. At first glance, I would expect sites with short vegetation to have a higher ET attenuation than sites with taller vegetation, especially because sites with shorter vegetation have a faster decline of ET (P 14 LL 4-5). The two results seem in contradiction. How do you explain this?

[Figure]

Is because ET in shorter vegetation is more coupled to GPP than to the decrease associated to soil water availability and this reflects in a lower value of d?

P. 17. LL 30. I would tend to disagree with this statement. The results show eventually that we need more eddy-covariance measurements everywhere or other type of observations that could be used for similar purposes. Overall, semiarid regions are more resilient to decay of ET according to Fig. 10.

References

Martin-StPaul, N., Delzon, S. & Cochard, H. Plant resistance to drought depends on timely stomatal closure. Ecol. Lett. 20, 1437–1447 (2017).

Leuning, R., 1995. A critical appraisal of a combined stomatal-photosynthesismodel for C3 plants. Plant. Cell Environ. 18 (4), 339–355

Ball, J., Woodrow, I., Berry, J., 1987. A model predicting stomatal conductance andits contribution to the control of photosynthesis under differentenvironmental conditions. In: Progress in Photosynthesis Research: Volume 4

---

## Referee Comment (RC3) · Anonymous Referee #3 · 2 Jan 2019

**Review of:**
**Boese et al., Carbon-water flux coupling under progressive drought**

The authors analyse dry-down periods at 31 flux tower sites to evaluate semi-empirical water use efficiency models. The authors present interesting concepts for analysing the effects of water stress on water use efficiency and separating mechanisms controlling vegetation drought responses. Whilst I appreciate the authors' efforts to understand deficiencies in the WUE models, the manuscript currently offers little process-level understanding, rather focusing on metrics and locally calibrated empirical models. It would be great to see the authors expand on their findings. For example:

1) The authors conclude that the WUE models need to consider radiation and soil water limitation of transpiration to better capture WUE (ET) changes during water stress, in addition to GPP and VPD effects. Yet, the authors offer very little in the way of explanation for why these limitations are important. What are the specific mechanisms? The other points in the discussion are rather obvious (short vegetation is generally more drought-prone and seasonal climates are more resistant to drought stress) but the main conclusion is spared little attention.

2) It would also be useful to unpack the results for "tall" vs "short" vegetation further. Whilst I acknowledge the limited metadata available for flux tower sites, the authors have potentially missed an opportunity to identify which ecosystems responds to water stress more strongly. For example, are sites with specific vegetation types, climates (hot, cold, wet, dry…?), high/low leaf area more responsive? Similarly, are the additional Rg and SWL terms more important in specific environments? "Tall" and "short" vegetation seems a rather simplistic classification to understand how specific sites responds to water stress and the broad vegetation classes are not particularly informative (for example, is savanna really "tall" vegetation or mainly grasses?).

3) I would question the meaning of the $S_{rem}$ variable. This measure doesn't account for antecedent conditions, which could play a large role in determining the rate of dry-down, especially during the short-term droughts analysed here. It is thus unclear what specifically can be learnt from the inclusion of this term? The authors might be able to test the sensitivity of their results to the assumption that antecedent conditions are negligible by e.g. calculating rainfall accumulation prior to the dry-down. I also note much of the manuscript discusses water use efficiency, yet all the equations and results are for ET?

Overall, the authors have provided a comprehensive and well-written, but rather superficial analysis of ET responses to water stress. It is not clear how the results can be used more widely to gain mechanistic understanding of ecosystem functioning under water stress, or improve the formulation of these processes in models, as they rely primarily on locally-calibrated statistical models. I would encourage the authors to dedicate less space on metrics and calibration schemes (some of this might be better suited to the supplementary?), and unpack their findings further.

**Specific comments:**

P1 L19: prime-sources should be primary sources?

P1 L22: interacting rather than interlocking?

P3 L3: Please correct spelling to La Thuile

P3 L8: Please specify what you mean by "the established methods"?

P3 L12: How did you define a precipitation event (> 0mm?)?

P3 L15: How did you handle observed vs. gap-filled data? If some of the dry-down periods were heavily gap-filled or missing, were these still analysed? If so, I would question what can be learnt from these sites as it seems unlikely the gap-filled data can accurately reflect fluxes during extreme conditions. Also how were the sites selected? On line L22 you mention 31 sites were used, but there are many more in the La Thuile release alone (of course not all with dry-downs). I'm surprised if there are only 47 dry-down events in the 200+ site records, but this is of course possible.

P3 L19: $Rn$ not defined? Also, EF is normally defined as $Qle/(Qle+Qh)$, with the latter part equating to $Rn – G$

P3 L20: Because the definition of dry-down events is central to this manuscript, I would like to see more details presented here instead of the supplementary. Also suggest using "we" instead of "I" in the supplementary.

P3 L25: remove data-sets

P4 L2: Mass balance is also affected by output from runoff. Similarly in the following sentence, stored water can depend on subsurface runoff. It seems reasonable to assume these fluxes were small due to the lack of precipitation, but this should be mentioned

P4 L10: How many missing values did you allow for?

P4 L19: How was uWUE determined?

P5 Eq. 5: Should $max(S_{remt})$ be without the $t$ subscript? Surely $S_{remt}$ is a single value, at time $t$?

P5 L15: Compared instead of inverted?

P6 L2: Consider mentioning that 1.0 is the best possible MEF value for readers not familiar with the metric

P9 Eq. 14: What is $ET_{rad}$?

P9 L29: Remove "was that"

P15 L 6: Correct to "led"

All figures: Avoid analysing results in the figure captions (e.g. models underestimated…), the captions should merely explain what is shown in the figures.

Fig 6: It is hard to see the blue line especially

---

## Author Comment (AC1) · 15 Feb 2019

We thank Martin deKauwe for his valuable feedback on the submitted manuscript. Below, we address general remarks and important specific remarks that required a response and describe how we incorporate these in the revised manuscript. In addition we carefully considered all specific comments related to spelling, clarity and references and integrated them into the revised manuscript where appropriate.

GENERAL REMARKS

- "In particular, I was hoping to learn about differences in dry- down as a function of vegetation types, hydro-climate, frequency of droughts, etc?".

This point raises an important issue. We also considered a more granular analysis of

underlying site-properties that could potentially explain the observed variability of obtained metrics. However, the limited sample size of this study did not allow for detailed stratifications of the data set. Nevertheless, in the revised manuscript, we provide a better presentation of how the results can be disentangled according to climate and vegetation types. To account for the small sample size, we now aggregated multiple climate types (tropical, mediterranean, temperate-humid) and vegetation types (short with grasslands and crops, mixed with savannas and tall for forests). However, we agree that it would be ideal to ultimately link the observed patterns to the physical properties of the plants rather than ecosystem-scale proxy variables. This is an important point that is now stressed in the discussion.

- "'We also explored [...] hydro-climatic properties of the sites' and I don't really see where they've done this?"

We agree that the previous wording failed to connect this statement to results presented later in the manuscript. Specifically, we refered to the mean seasonal WAI amplitude as indicator for regularly occurring water-limitation. We have clarified the manuscript accordingly.

- "Finally, I'm not clear why the authors only focus on ET? The paper frames the question around WUE and so they should also look at the evolution of GPP during a drydown, shouldn't they? They could easily argue that GPP isn't directly observed and that is fine, but then I think changing the framing more clearly towards ET only, including removing "carbon" from the title, is warranted."

We agree that the previous version of the manuscript failed to convey a central part of how our analysis was conceived. As also remarked by Reviewers 2 & 3, there is a discrepancy between the stated goal of examing carbon–water coupling via water-use efficiency models and the fact that most of the analysis take transpiration as the target variable. Here, we do not assume that the measured gross primary productivity exhibits any less observational and processing uncertainties. In brief, using ET = f(GPP, x)

instead of WUE = f(x) is merely a reformulation that focusses on how different WUE models affect the flux magnitudes of ET rather than the ratio WUE = GPP / ET. In the latter approach, small GPP and even smaller ET values can lead to very high WUE values and can in a least-squares regression bias the analysis towards time periods that should not receive as much weight. We have thus added an appropriate paragraph to the introduction.

- "The authors propose the need for two additional corrections, one related to radiation and the other soil water availability. I've commented on this below, it feels unnecessary (mechanistically) and a form of an artificial correction, but I'm happy to be corrected on this and keen to read a more thorough justification."

This comment engages a critical part of our analysis. For our level of analysis, we used a semi-empirical approach, the definition of which we also have explain more prominently in the revised manuscript. The approach is then primarily guided by empirical criteria such as goodness-of-fit measures, while aiming at effective model structures that can be related to physical processes at aggregated scales. In previous work, this approach was used by Boese et al. (2017) to identify a previously neglected driving effect of radiation on transpiration. As we also lay out in Fig. 1, the radiation-effect itself is beneficial to model performance both outside and inside dry-down events. Yet its inclusion exacerbates systematic model errors (Fig. 2), which in turn require correction. The chosen approach is thus primarily motivated by empirical performance of the models. Yet while we succeeded in remediating the model performance during dry-down events, the link to responsible mechanisms does indeed remain tenuous. In the revised manuscript, we discuss this problem in more depth.

- "A number of studies [...] have highlighted the need for a non-stomatal correction to GPP (which indirectly affects ET) in order to correctly capture observed responses. This isn't commented on here, but I note that the authors seem to be arguing the opposite, that is, there is a need for a more direct correction on ET but that GPP is fine."

This is a valuable idea to discuss. In the previous version of the manuscript, we did not assume any non-stomatal limitations of GPP during water-limitation. It is nevertheless important to consider to which degree our analysis, if implicitly, addressed this point. The model Zhou+SWL predicts ET as a function of both GPP and soil-water limitation. In our conceptualization, the +SWL term serves as a corrective for non-stomatal limitations of ET. Yet it would also be possible to see the term as correcting for any difference in how soil-water limitation affects ET vs. GPP. Nevertheless, this is an important complication that deserves more attention in the discussion.

- "I really think it is important that the authors document all their fitted terms, e.g. the terms in the supplementary, otherwise this study isn't reproducible."

Agreed. We added the values of the optimized parameters as table in the supplement.

- "What role does LAI, or more specifically, leaf turnover play in the modelling done here? Is it possible that some events see leaf area adjustments which could impact on ET fluxes?"

We agree that changes of LAI have been neglected until now. Especially for dry-down events in vegetation adapted to humid conditions, decreasing LAI due to drought stress has been observed (Anderson et al. 2015). For our purpose, we would expect any negative change in LAI to both affect ET and GPP negatively, as both fluxes depend on the effective surface area at which carbon uptake and water loss happen. It thus seems probable that changes in LAI would not manifest in changing WUE during drought.

SPECIFIC COMMENTS AND CRITICISM

Abstract

- "As written, I feel like it requires a fair amount of prior insight to follow [...]"

We edited the abstract to be more informative and easy to understand for readers unfamiliar with our approach.

Introduction

- Pg 1, line 20: it would be nice (but optional) to have a few physiological citations alongside the point about GPP decline with water limitations.

We have added appropriate citations at the respective location.

- Pg 2, line 18: the text about atmospheric and soil droughts co-occurring ... It reads as if there is an alternative? Surely, as far as the vegetation is concerned these two will always co-occur? If there is plenty of soil water, then even if there is a precipitation drought, it is not a drought for the vegetation. Am I missing something? I assume the point that is being made here is for the need to seperate out the response to VPD vs the response to soil water. I think this could be more clearly articulated here.

This was indeed the point and we have clarified the text accordingly.

Methods

- Pg 3, line 24 onwards: this text isn't clear enough - "namely the quantity does not necessarily reflect the water-stress actually experienced by the plants" - what specifically do the authors mean? Do they mean because these data are usually of limited depth, so do not fully reflect the root- zone?

Thank you for pointing this out. Yes, partially because of differences between rooting-depth and the depth of soil-water measurements. But also because the soil-water contents at specific depths would need to be weighted with the root water uptake which can differ substantially based on root architecture and physiology (Schneider et al. 2010).

- Pg 4, Eqn 1: What about groundwater? This deserves some mention here, if only to highlight it in the assumptions made.

This is correct, we now state that we make the assumption that this does not include groundwater access.

- Pg 4, line 20: Again ... the text about the Boese study and radiation requires further explanation. I suggest it is done once and then it could be referred to as done here. I need to read this paper, but my initial reaction is to query the statement. Why is radiation an important driver of transpiration, independent of GPP? And why Rg and not net radiation? This feels like a form of double counting here (radiation via PAR is a driver of GPP and Rnet is a driver of ET)? Clarifying this in the text would be worthwhile for the reader.

We agree that the manuscript assumed too much knowledge regarding the study of Boese et al. (2017). In that study, the authors identified that an additional radiation term was necessary to predict ET from GPP and VPD at the ecosystem-scale. Similar to the present study, this finding was thus an empirical one, justified by the performance of the models at multiple sites in cross-validation. Yet this finding can be connected to the theory of Jarvis and McNaughton (1986), in which one part of transpiration is driven by the gradient (imposed transpiration, in our case GPPÂůVPDˆ0.5) and the other is driven by the radiative energy input (equilibrium transpiration, in our case r * Rg). While preparing the analysis of the impact of radiation on WUE, we also considered Rnet. As the model performance was slightly higher for Rg and as both variables are temporally very strongly correlated for each particular site, we used Rg in that study. However, this is merely one possible explanation discussed in the preceding publication for what is an empirical pattern. We acknowledge that this needs to be clarified for readers not familiar with that work.

- On a related point - what about evidence of the need for a non-stomatal limitation of photosynthesis during drought? How do the authors suggest this factors into their analysis?

We addressed the closely related point regarding non-stomatal limitations of GPP above in the section "General Remarks".

- Pg 5, Line 1 onwards: "Both models"?? I assume the authors mean eqn 3 and 4? It

isn't clear. I don't follow this text - the soil water availability would also have an effect on ET, the reduction in stomatal conductance due to drought would lead to reduced ET. The text as written makes it appear that this only affects GPP. They then propose an empirical correction on transpiration for declining soil water. I fail to see why this is necessary? The ET quantity reflects the soil water availability? I find this quite worrisome, as above with the Rg, this feels like a double correction that isn't warranted mechanistically.

We agree that the paragraph is unclear and can be misunderstood. In fact, we just wanted to state that while the models of eq. 3 & 4 (clarified in the revised version) do not contain an explicit variable for soil-water limitation, one can assume that any decrease of stomatal conductance would lead to reductions in GPP. As ET is here predicted from the variables on the right-hand side, any reduction of GPP induced by water-limitation would entail reductions in ET. The mentioned reductions introduced with the +SWL variants are necessary as Fig. 2 and especially Fig. 3 suggest that models with constant uWUE and r parameters fail to predict ET acurately over the course of dry-down events. More mechanistically, the introduction of the s factor in eq. 6 could be seen as fulfilling a function similar to $g\_1$ attenuation of stomatal conductance models in response to water-limitation.

- Pg 5, eqn 5: where is q given by site? It needs to be shown to the reader.

This has been added to the supplement together with the other fitted parameters.

- "Short" vs "Tall" feels a pretty vague distinction. I think the tall category would have considerable variability and it would be more interesting to consider the results in the context of the actual heights rather than this arbitrary binary classification. I am aware that it is difficult to obtain these kinds of site characteristics, so the authors do not need to do this; however, I think it would be more interesting if they could.

In the updated version of the manuscript, we added a third category, "mixed", for sa-vannah type ecosystems. This admittedly only partially resolves the problem that vege-

tation types are only crude proxies for the actual height of plants in ecosystems (which in turn can vary substantially for any given site). However, we also clarify that the stratification can reflect – through predominating growth forms – both differences in water-use strategies and rooting depths. Yet it has to be stated that these categories are at best imperfect proxies for variables that as of now are not at all or not consistently measured.

- Pg 8, Eqn 11: how does k vary between sites?

We apologize for the omission. For this analysis, we fixed k at 0.05 which is a reasonable expectation on a global scale (Teuling et al. 2006, also added the appropriate citation in the manuscript).

- How sensitive are the results from eqn 11 to the assumption of a WAI of 100 mm?

To address this point, we reran the analysis with three different values of $WAI_{max}$ (as now referred to in the manuscript): 70, 100 and 130 mm. The corresponding plots with labels of the IGBP vegetation classes are attached below. The results suggest that there is indeed some sensitivity of our results, yet all levels show a significant correlation between k and the seasonal amplitude of dryness (higher correlation for lower $WAI_{max}$).

Results

- does figure 2 need to be a figure? It strikes me that it could as easily be a table? It might be then preferable to give an example of a time-series between each model evaluation?

We think that Fig. 2 is useful as it visually represents the fundamental motivation of the study: Namely that both the Zhou and +Rg models fail to predict acurately during periods of water-limitation. However, we agree that the importance of this discrepancy has not been properly addressed in the manuscript itself. We would prefer this as a figure, as the variability inside the groups (95% CI intervals) can not be easily rendered

in text. We further concur that time-series can be helpful to understand the model errors. While Fig. 3 averages the time-series of multiple sites, we added instructive examples of individual sites in the supplement.

- Pg 10, line 10: This point about the ET not declining fast enough, would fit with the narrative I presented earlier of the need for a non-stomatal limitation on GPP, which would also reduce gs and so ET. Of course this wouldn't work for this kind of empirical model. The correction (SWL) could be seen as effectively doing this, although I don't follow the justification for this approach.

This is an interesting point. In our model, the attenuating factor s could be seen as reflecting possible – process-agnostic – differences in the drought-sensitivity of GPP vs. ET. If, for example, GPP is additionally limited by non-stomatal factors during water-limitation, our model would be expected to underpredict ET (which is still mostly limited by stomatal conductance). If the +Rg+SWL model instead overestimates ET for longer events – while observed ET declines faster – it suggests that ET is more limited by non-stomatal factors when compared to GPP. It is however important to stress in the discussion that due to the empirical nature of the approach, observed patterns can only be tenuously be mapped back to particular processes.

- The interpretation of figure 8 seems a bit optimistic and at the very least should be justified ("significant association") with statistics.

Agreed. We referred to the confidence interval of the local polynomial regression used for smoothing. However, the same statement can be better supported with a linear model, which we use in the updated version of the manuscript.

- Figure 9 ... the difference in k is presented in terms of the "height" of the vegetation, whereas in my eyes it could as easily be interpreted as related to rooting depth and/or leaf area. I'd suggest that height as an explanatory of the difference in dry-down doesn't really have a mechanistic interpretation. At the very least the authors should outline what they think it is a proxy for, or state more clearly how height impacts on the rate of

dry-down? Are they hypothesising it is via differences in roughness length?

The tall/short distinction can indeed be seen as an approximate indicator for both water-use strategies and mean rooting depths (see longer response above). We did not consider the distinction to be mediated by differences in roughness length.

Discussion

- Pg 15, Line 10: "Our analysis suggests an ecosystem scale soil- water availability effect on WUE that is statistically independent from VPD effects on the contraction of stomata" - This is a big claim, where is this supported in the data, it would be really helpful to link this to the results. Furthermore, the authors need to unpick this further. If it is independent of the response of gs to VPD, can they discuss the mechanisms they are invoking, presumably via the soil water. Why would it be invariant across ecosystems? This would argue against much of the emerging plant hydraulics literature, surely? Or have I simply misunderstood? I actually see they then link this to a hydraulic limitation related to height - which begs the need to be far more detailed in this analysis. In my eyes it is not sufficient to arbitrarily split the vegetation into small and tall and then to invoke a hydraulic explanation. The tall category could conceivably include a range of heights, do the authors know for certain it is largely made up of very tall trees? I am concerned this is pretty speculative to be honest.

This is a critical part of our discussion and we agree this needs to be discussed more carefully and linked better to the results. The statistical VPD-independence is connected to the observation that the Zhou-Model on its own cannot acurately predict the ET decline during dry-down events. This model integrates the mentioned effect of VPD on stomatal conductance (Zhou et al. 2015). As we demonstrate, this alone proves insufficient to explain ET decline during dry-down events (Fig. 2, 4). Yet even integrating the effect of soil-water limitation (Zhou+SWL) on uWUE (which is inversely proportional to $g\_1$) did not provide substantial benefits to model performance (Fig. 4). Instead the complete reduction of stomatal and non-stomatal ($r * Rg$) transpiration

components (+Rg+SWL) provided the highest performance of predicted ET. As the attenuating factor s is not exclusively reducing stomatal conductance in this model, it could be interpreted as sign of a process affecting both source of transpiration. A reduced stem hydraulic conductivity during water-limitation (Ladjal et al. 2005), could be responsible for this generalized decrease of transpiration. Nevertheless, our empirical approach at ecosystem-scale makes it difficult to pinpoint the mechanism responsible for the observed effects. In the discussion, we now make this clear and further highlight the importance of following up on the results with mechanistic studies in controlled settings.

Importantly, the reduction effect is certainly not invariant across ecosystems. As we show in Fig. 7, the effective reduction of ET varies notably between different ecosystems.

Additional References

Anderson, M. C., Zolin, C. A., Hain, C. R., Semmens, K., Yilmaz, M. T., & Gao, F. (2015). Comparison of satellite-derived LAI and precipitation anomalies over Brazil with a thermal infrared-based Evaporative Stress Index for 2003–2013. Journal of Hydrology, 526, 287-302.

Ladjal, M., Huc, R., & Ducrey, M. (2005). Drought effects on hydraulic conductivity and xylem vulnerability to embolism in diverse species and provenances of Mediterranean cedars. Tree physiology, 25(9), 1109-1117.

Schneider, C. L., Attinger, S., Delfs, J. O., & Hildebrandt, A. (2010). Implementing small scale processes at the soil-plant interface–the role of root architectures for calculating root water uptake profiles. Hydrology and Earth System Sciences, 14(2), 279-289.

Zhou, S., Yu, B., Huang, Y., & Wang, G. (2015). Daily underlying water use efficiency for AmeriFlux sites. Journal of Geophysical Research: Biogeosciences, 120(5), 887-902.

[Figure]

[Figure]

Fig.: Response of the relationship of k to the amplitude of seasonal dryness for three different values of $WAI_{max}$.

**Fig. 1.**

[Figure]

Fig.: Sensitivity of the comparison of predicted vs. observed k for three different calculations of $S_{rem}$. (a) Using the upper bound of the 95% confidence interval of the calculation of the initial $S_{rem}$, (b) the most likely value of the initial $S_{rem}$, as used in the manuscript, (c) using the lower bound of the 95% confidence interval.

**Fig. 2.**

[Figure]

Fig.: Sensitivity of the comparison of model performances for three different calculations of $S_{rem}$. (a) Using the upper bound of the 95% confidence interval of the calculation of the initial $S_{rem}$, (b) the most likely value of the initial $S_{rem}$, as used in the manuscript, (c) using the lower bound of the 95% confidence interval.

**Fig. 3.**

[Figure]

---

## Author Comment (AC2) · 15 Feb 2019

We thank Referee #2 for his valuable feedback on the submitted manuscript. Below, we address general remarks and important specific remarks that required a response and describe how we incorporate these in the revised manuscript. In addition we carefully considered all specific comments related to spelling, clarity and references and integrated them into the revised manuscript where appropriate.

GENERAL REMARKS

- "This is partially recognized in the final Section 4.3 (P. 18 LL 7-8) but I have the feeling that some of the statement about the utility of this metric (e.g., P 18 LL 14-17) could be overoptimistic due to the local calibration and strong variability across sites."

[Figure]

We agree that the discussion of our results requires a better differentiation between the diagnostic insights we could provide and its applications in future work. Due to the limitation of this metric (specifically the local calibration and its recursive, ET-dependent character), any future work should first try to link it with directly measurable variables. If representations of such variables reflecting soil-water content can be derived from mechanistic land surface models, this could provide an additional step to verify the presented patterns in ET–GPP coupling. We have changed the discussion accordingly.

- "[...] This scatter is probably due to observational uncertainties but also to behavior of the different ecosystems in response to specific dry-down events and to the definition of S_rem (see below). A few additional words on this problem could be added."

The limited sample size of the study and vast variability in sampled climate, plant and ecosystem types does in fact pose a substantial challenge for obtaining generalizable understanding with a small sample of sites. We have added this to the section discussing limitations of our approach.

- "There is not a representation of how WUE (e.g., GPP/ET) varies with Srem based on observations."

We concur that such a representation could be helpful to understand to what degree GPP–ET coupling holds during the periods of interest. We propose for this purpose to further not only show the covariation of Srem with WUE but also uWUE as proposed by Zhou et al. (2014 & 2015), which already accounts for the dependence of water-vapor diffusion and stomatal conductance on VPD.

- "However, the main issue I have with the definition of Srem is that it cannot keep track of any precedent effect of water availability or soil water stress in the system." \

This is an important point worth discussing more extensively in the manuscript. The Srem metric and its analysis is certainly limited to the approximately exponentially decaying ET of dry-down events. However, antecedent conditions can be reflected in this

metric. Consider an ecosystem that experienced intermittent periods of water-limitation that did not qualify as dry-down events according to our definition. After a given last, weak precipitation event, we might see a longer period without any rain-fall in which ET starts following an exponential dry-down decay. Even though we identify only the latter part as dry-down event, plants in the ecosystem are already in drought stress at the beginning of the event. Yet this lower water availability would then also manifest in the reduced ET at the beginning of the event and subsequently a smaller integral of ET used to obtain Srem. Any normalization of the variable for the sites (p4 l10–11) will of course prevent a possible interpretation of q values between sites (as Srem_max can no longer be compared across sites).

We further verified the robustness of our results by using two additional calculations of Srem. In these, we used the lower and upper 95% confidence intervals of the parameters (ET_0 and k) used in the exponential model to obtain a higher and lower variant of Srem. The discrepancy of the two Srem calculations therefore incorporates uncertainties about how the exponential fit could capture the ET decline despite unknown initial conditions and missing values in the time series. We attached this comparison as figure below. Nevertheless, antecedent conditions might well be responsible for deviations of the highly idealized behavior of the models we employed. As such, we haven given this limitation more prominence in the discussion.

SPECIFIC COMMENTS

- "P 2. LL 7. As a matter of fact, stomatal closure is occurring always at higher potentials than critical cavitation levels for xylem (Martn-StPaul et al 2017)."

We have amended the sentence in question.

- "P. 2 LL 8. Increased leaf-temperature does not necessarily lead to a decrease in photo- synthesis; it depends on the actual temperature and temperature-sensitivity of a given species."

We agree that this statement was too generalized. We have corrected this in the revised manuscript.

- "P. 2. LL 25. Why are you stating that ET and soil moisture are following a linear relation? Is this following the exponential decrease of ET with time? Then, very likely, the linearity is with some "proxy" values of soil moisture as S_rem and not with the actual soil moisture."

For our analysis, we assumed that the rate of supply-limited ET depended linearly on the water available for root water uptake (Teuling et al. 2006) as in a one-storage water balance model. Thus: $ET \sim k * Srem$. We agree that even this simplification only holds for the plant-available water and not for the total soil moisture. We have corrected the sentence accordingly!

- "P. 8 LL 10-17. For how many steps the WAI_t variable is computed? Since the beginning is from the arbitrary 100 mm in order to extract the mean seasonal cycle of WAI, you need several years."

This is correct. We used 115 years from CRUNCEP reanalysis to obtain mean seasonal WAI amplitudes. Further, we reran the analysis with three different values of WAI_max (as now referred to in the manuscript): 70, 100 and 130 mm. The corresponding plots with labels of the IGBP vegetation classes are attached below. The results suggest that there is indeed some sensitivity of our results, yet all levels show a significant correlation between k and the seasonal amplitude of dryness (higher correlation for lower WAI_max).

- "P. 8. LL 19. Given how WAI is computed, memory effects refer only to seasonal effects, since WAI is averaged."

We have clarified this in the description of the metric.

- "P. 9. LL 21. This is very much expected since they do not have any way of accounting for soil-water limitations."

This is an important point. The two models indeed do not contain explicit variables reflecting the soil-water status of the ecosystems. Yet indirectly, observed reductions of GPP, even with a constant underlying water-use efficiency uWUE, could reflect plant responses to soil-water scarcity. Yet we agree that this wasn't phrased well enough and have amended the sentence accordingly.

- "P 13. LL 11-12. This result is a bit counterintuitive to me. At first glance, I would expect sites with short vegetation to have a higher ET attenuation than sites with taller vegetation, especially because sites with shorter vegetation have a faster decline of ET (P 14 LL 4-5). The two results seem in contradiction. How do you explain this? Is because ET in shorter vegetation is more coupled to GPP than to the decrease associated to soil water availability and this reflects in a lower value of d?"

Thank you for noting this crucial point. We believe that the two observations do not have to be seen as standing in contradiction. As you mention, in low vegetation types (in our case dominated by grasslands), rapidly declining GPP seems to be largely sufficient to predict the also quickly diminishing ET. For tall vegetation types (dominated by trees), more gradual, possibly hydraulic limitations could lead to a shallower decline of ET, while a deeper root zone can sustain ET for comparatively longer periods.

- "P. 17. LL 30. I would tend to disagree with this statement. The results show eventually that we need more eddy-covariance measurements everywhere or other type of observations that could be used for similar purposes. Overall, semiarid regions are more resilient to decay of ET according to Fig. 10."

We concur that this statement does not follow from our results in its current form. What is primarily needed is more focus on regions prone to intermittent water-scarcity. For semi-arid regions, it is better to see the remaining scatter rather than than mere amplitudes as indication for more eddy-covariance measurements. We have clarified this in the discussion.

Additional References

Teuling, A. J., Seneviratne, S. I., Williams, C., & Troch, P. A. (2006). Observed timescales of evapotranspiration response to soil moisture. Geophysical Research Letters, 33(23).
* * *
[Figure]

[Figure]

Fig.: Response of the relationship of k to the amplitude of seasonal dryness for three different values of $WAI_{max}$.

**Fig. 1.**

[Figure]

Fig.: Sensitivity of the comparison of predicted vs. observed k for three different calculations of $S_{rem}$. (a) Using the upper bound of the 95% confidence interval of the calculation of the initial $S_{rem}$, (b) the most likely value of the initial $S_{rem}$, as used in the manuscript, (c) using the lower bound of the 95% confidence interval.

**Fig. 2.**

[Figure]

Fig.: Sensitivity of the comparison of model performances for three different calculations of $S_{rem}$. (a) Using the upper bound of the 95% confidence interval of the calculation of the initial $S_{rem}$, (b) the most likely value of the initial $S_{rem}$, as used in the manuscript, (c) using the lower bound of the 95% confidence interval.

**Fig. 3.**

---

## Author Comment (AC3) · 15 Feb 2019

We thank Referee #3 for his valuable feedback on the submitted manuscript. Below, we address general remarks and important specific remarks that required a response and describe how we incorporate these in the revised manuscript. In addition we carefully considered all specific comments related to spelling, clarity and references and integrated them into the revised manuscript where appropriate.

GENERAL REMARKS

1. MECHANISMS OF LIMITATION

This is a critical point of our approach and we agree that the previous version of the manuscript communicated this insufficiently. In a previous study, Boese et al. (2017)

first observed the existence of a GPP-independent association of transpiration to radiation. As in the present work, its semi-empirical approach targeted at a high model performance of predicted evapotranspiration, while associating the detected effects to plausible physical variables. In our manuscript, we aimed to expand this research to water-limited periods, in which the water supply is an additional factor controlling transpiration rates.

As we demonstrate, declining GPP due to stomatal contraction proves insufficient to explain ET decline during dry-down events (Fig. 2, 3, 4). Yet even integrating the effect of soil-water limitation (Zhou+SWL) on uWUE did not provide substantial benefits to model performance (Fig. 4). Instead the complete reduction of stomatal and non-stomatal (r * Rg) transpiration components (+Rg+SWL) provided the highest performance of predicted ET. As the attenuating factor s is not exclusively reducing stomatal conductance in this model, it could be interpreted as sign of a process affecting both source of transpiration. A reduced stem hydraulic conductivity during water-limitation (Ladjal et al. 2005), could be responsible for this generalized decrease of transpiration. Nevertheless, our empirical approach at ecosystem-scale makes it difficult to pinpoint the mechanism responsible for the observed effects. In the discussion, we now make this clear and further highlight the importance of following up on the results with mechanistic studies in controlled settings.

2. STRATIFICATION OF SITES ALONG VEGETATION STRUCTURES

In the updated version of the manuscript, we added a third category, "mixed", for savannah type ecosystems. This admittedly only partially resolves the problem that vegetation types are only crude proxies for the actual height of plants in ecosystems (which in turn can vary substantially for any given site). However, we also clarify that the stratification can reflect – through predominating growth forms – both differences in water-use strategies and rooting depths. Yet it has to be stated in the manuscript that these categories are at best imperfect proxies for variables (e.g. average rooting depth or plant water-use strategies of woody vs. non-woody plants) that as of now are not

at all or not consistently measured. Overall, the semi-empirical models we employed provide an effective description of how different ecosystem fluxes interact empirically. While the calibration to local properties impedes ad-hoc generalizations, the variability of parameters between sites can be interpreted as reflecting the variability of ecosystem functional properties (such as uWUE). For ecosystems containing various plant types with differing structural or physiological properties, the observed patterns are then aggregated signals for the whole system.

**3. SREM AND ANTECEDENT CONDITIONS**

The utilized Srem variable does indeed have important shortcomings. As we described, our motivation for its introduction was to serve as a proxy variable for extractable soil-water that does not rely on incomplete and inconsistently measured observations of soil-water content. However, due to its nature as calculated proxy metric, it suffers some notable limitations. As you mention, its reliance on an approximately exponentially decreasing ET omits preceding water-stress. However, antecedent conditions can be reflected in this metric. Consider an ecosystem that experienced intermittent periods of water-limitation that did not qualify as dry-down events according to our definition. After a given last, weak precipitation event, we might see a longer period without any rain-fall in which ET starts following an exponential dry-down decay. Even though we identify only the latter part as dry-down event, plants in the ecosystem are already in drought stress at the beginning of the event. Yet this lower water availability would then also manifest in the reduced ET at the beginning of that event and subsequently a smaller integral of ET used to obtain Srem. Any normalization of the variable for the sites (p4 l10–11) will of course prevent a possible interpretation of q values between sites (as Srem_max can no longer be compared across sites).

We further verified the robustness of our results by using two additional calculations of Srem. In these, we used the lower and upper 95% confidence intervals of the parameters (ET_0 and k) used in the exponential model to obtain a higher and lower variant of Srem. The discrepancy of the two Srem calculations therefore incorporates uncertainties about how the exponential fit could capture the ET decline despite unknown initial conditions and missing values in the time series. We attached this comparison as figure below. Nevertheless, antecedent conditions might well be responsible for deviations of the highly idealized behavior of the models we employed. As such, we haven given this limitation more prominence in the discussion.

**3.1 RELIANCE ON LOCALLY-CALIBRATED STATISTICAL MODELS**

The local optimization of parameter values is certainly a limitation if the insights are to be generalized or included in mechanistic models. The lack of firm process understanding on the scale of ecosystems does however make a semi-empirical approach a valuable approach to capitalize on the availability of eddy-covariance observations for whole ecosystems. In this approach, local parameter estimates are understood as ecosystem functional properties which regulate ecosystem responses to environmental conditions. Therefore, the empirical justification of model terms (such as the linear Rg-term and the +SWL term) and systematic patterns in their parameter estimates provide information about the interaction of variables on ecosystem scale. Nevertheless, we agree that this decidedly non-mechanistic approach has shortcomings that more process-motivated investigations can address. For the purpose of the study, we see the detection of the soil-water limitation effect and its variability across sites as a good starting point for further work. To clarify which mechanisms might be responsible for the effect and how they drive differences between ecosystems, different observations such as leaf and xylem water potentials as well as volumetric soil-water content might be necessary. In the revised introduction and discussion, we provide a better explanation for the scope of our study and highlight how our findings could stimulate experiments under controlled conditions and factorial model experiments. We also omitted the separation into two different calibration schemes which unnecessarily distracts from the main outcomes of the analyses.

**3.2 WATER-USE EFFICIENCY VS ET IN EQUATIONS**

We agree that our choice of ET as metric to evaluate water-use efficiency models needs a better explanation. In brief, using ET = f(GPP, x) instead of WUE = f(x) is merely a reformulation that focusses on how different WUE models affect the flux magnitudes of ET rather than the ratio WUE = GPP / ET. In the latter approach, small GPP and even smaller ET values can lead to very high WUE values and can in a least-squares regression bias the analysis towards time periods that should not receive as much weight. We have thus added an appropriate paragraph to the introduction.

SPECIFIC REMARKS

- "P3 L12: How did you define a precipitation event (> 0mm?)?"

We used a cut-off value of 0.2 mm/d to define precipitation events. We have added this criterion to the methods section.

- P3 L15: How did you handle observed vs. gap-filled data? If some of the dry-down periods were heavily gap-filled or missing, were these still analysed? If so, I would question what can be learnt from these sites as it seems unlikely the gap-filled data can accurately reflect fluxes during extreme conditions. Also how were the sites selected? On line L22 you mention 31 sites were used, but there are many more in the La Thuile release alone (of course not all with dry-downs). I'm surprised if there are only 47 dry-down events in the 200+ site records, but this is of course possible.

Thank you for highlighting this important point. In the selection and preprocessing of the data, we applied strict filtering that only uses high-quality data that is either observed directly or gap-filled with high confidence. If this filtering resulted in gaps during dry-down events, they were only filled by interpolated values from an exponential fit to allow the calculation of a continuous time series of Srem. In all model fitting and evaluation, days with low quality observations in one variable were omitted completely from the analyses. Nevertheless, the fact that dry-down events, when not occurring seasonally, represent extreme conditions where data quality becomes particularly important is now stated explicitly in the revised manuscript.

- "P4 L10: How many missing values did you allow for?"

We did not set a specific threshold value for missing values during dry-down events. Instead, we check whether an exponential model could explain at least 40% of the variability of ET during these events. Please also see our comment regarding the antecedent conditions above for how we ascertained that the uncertainties in parameter estimations originating from longer gaps did not affect our results qualitatively.

Additional References

Boese, S., Jung, M., Carvalhais, N., & Reichstein, M. (2017). The importance of radiation for semiempirical water-use efficiency models. Biogeosciences (Online), 14(12).

Ladjal, M., Huc, R., & Ducrey, M. (2005). Drought effects on hydraulic conductivity and xylem vulnerability to embolism in diverse species and provenances of Mediterranean cedars. Tree physiology, 25(9), 1109-1117.

[Figure]

Fig.: Response of the relationship of k to the amplitude of seasonal dryness for three different values of $WAI_{max}$.

**Fig. 1.**

[Figure]

Fig.: Sensitivity of the comparison of predicted vs. observed k for three different calculations of $S_{rem}$. (a) Using the upper bound of the 95% confidence interval of the calculation of the initial $S_{rem}$, (b) the most likely value of the initial $S_{rem}$, as used in the manuscript, (c) using the lower bound of the 95% confidence interval.

**Fig. 2.**

[Figure]

[Figure]

Fig.: Sensitivity of the comparison of model performances for three different calculations of $S_{rem}$. (a) Using the upper bound of the 95% confidence interval of the calculation of the initial $S_{rem}$, (b) the most likely value of the initial $S_{rem}$, as used in the manuscript, (c) using the lower bound of the 95% confidence interval.

**Fig. 3.**

---

## Referee Report (RR1)

**Review of first revision of Boese et al., Carbon-water flux coupling under progressive drought**

Overall the authors have addressed the comments well. However, I do not think the use of ET is still sufficiently explained in the manuscript, given the framing around WUE in the abstract and introduction. The authors have added further justification and now state:

*"We use a large global archive of flux tower observations [...] to scrutinize water-use efficiency formulations during periods of increasing water limitation. To test the different models, we 35 evaluated them against day-time ET observations. This has the advantage that the absolute flux magnitudes of ET and GPP are taken into account."*

I agree that there are numerical issues with using WUE when ET or GPP values are low. Nevertheless, I do not think this section is clear enough nor explains to the reader why only ET is used in instead of WUE and GPP (albeit it is not directly measured). Ideally the authors should explicitly explain in this section why ET is used over WUE to avoid confusion. Also is it appropriate to equate ET with GPP, when the two can be decoupled under certain circumstances (e.g. https://onlinelibrary.wiley.com/doi/full/10.1111/gcb.14037)?

I will also note that the authors have only addressed 3 specific comments that I provided in my first review (out of 20 or so). I have provided some additional comments below but will not review the manuscript in detail again as I'm finding myself repeating previous comments that have not been addressed either in the manuscript or the reply.

P1 L11: Attenuation of what? Also suggest rewording "for all included FLUXNET sites" as "for *n* FLUXNET sites"

P2 L1: "due to photolimitation of photosynthesis". Why is this the main mechanism given light should be more ample during droughts (reduced cloudiness)?

P3 L8: Correct spelling is "La Thuile"

P6 L11: "In the absence of a knowning", please correct.

Finally, I would ask the authors to keep in mind that many scientists are female, and as such thanking anonymous reviewers for "his" comments should preferably be avoided.

---

## Author Response (AR2)

**Author's Response**

The manuscript has been revised according to the new and remaining comments by Referee #3. We are happy that the manuscript could be improved and that critical points regarding the choice of the target variable of our analyses have been expanded, thus making the motivation of our analyses clearer.

Sven Boese on behalf of the authors

**Reply to Referee #3**

We thank Referee #3 for the constructive comments on our manuscript "Carbon–Water Coupling Under Progressive Drought". We are grateful for the acknowledgement of the improvements made in response to the last round of reviewer comments. We apologize for the comments we either missed to integrate or which we failed to highlight in the manuscript. Below, we address new and remaining issues; the corresponding changes have been highlighted in green in the revised manuscript.

Thank you for noting the use of the male pronoun in our replies, we apologize for this!

**ET vs WUE as Target Variable**

Regarding the choice of evaluating the WUE models against ET rather than WUE itself, Reviewer #3 states:

> However, I do not think the use of ET is still sufficiently explained in the manuscript, given the framing around WUE in the abstract and introduction.

We agree that this point still requires a better clarification of our reasoning behind the choice of the target variable. We have thus amended this section in the introduction substantially, providing a better argumentation for choosing ET as target variable over WUE.

**Specific Comments of Round 2**

*P1 L11: Attenuation of what? Also suggest rewording "for all included FLUXNET sites" as "for n FLUXNET sites".*

We agree that this was unclear. We have added that this attenuation refers to ET and have added the number of included FLUXNET sites, which we also think is important for the abstract.

*P2 L1: "due to photolimitation of photosynthesis". Why is this the main mechanism given light should be more ample during droughts (reduced cloudiness)?*

This was erroneous and has been corrected to "limited photosynthesis".

*P3 L8: Correct spelling is "La Thuile" P6 L11: "In the absence of a knowning", please correct.*

Both cases have been corrected.

**Remaining Comments From the First Round of Reviews**

*P1 L19: prime-sources should be primary sources? P1 L22: interacting rather than interlocking? P3 L3: Please correct spelling to La Thuile*

All three suggestions/corrections have been integrated into the revised manuscript.

*P3 L8: Please specify what you mean by "the established methods"?*

We have clarified this part by referring to the paper of Papale et al. (2006) which outlines the established methods of quality-assurance and quality-control used for eddy-covariance observations.

*Also how were the sites selected? On line L22 you mention 31 sites were used, but there are many more in the La Thuile release alone (of course not all with dry-downs). I'm surprised if there are only 47 dry-down events in the 200+ site records, but this is of course possible.*

Indeed only a very small subset of EC-sites was used in our analysis. This is the result of the strict filtering for sufficiently long rain-free periods with decreasing ET/Rn and approximately exponentially decreasing ET. One possible way to circumvent this limitation would be to rely on alternative metrics quantifying the soil-water status, i.e. ones that would not require the integration over the exponential function to estimate the total remaining water storage. However, these metrics would either be mere climatological indicators of dryness (thus unable to incorporate the information available in the observed decline of ET) or consistent data sets of soil water observations for whole profiles and multiple sites (so far not available).

> *P3 L19: Rn not defined? Also, EF is normally defined as Qle/(Qle+Qh), with the latter part equating to Rn − G.*

This was indeed misleading. We have edited the point to note that we used the ratio of ET / Rn as indicator for supply- or demand-controlled evapotranspiration and no longer refer to it as evaporative fraction.

> *P3 L25: remove data-sets*

Done.

> *P4 L2: Mass balance is also affected by output from runoff. Similarly in the following sentence, stored water can depend on subsurface runoff. It seems reasonable to assume these fluxes were small due to the lack of precipitation, but this should be mentioned.*

This has been amended in the revised version.

> *P4 L19: How was uWUE determined?*

We now clarify that uWUE is an empirical parameter that can be estimated using statistical regression techniques.

> *P5 Eq. 5: Should max(Srem_t) be without the t subscript? Surely Srem_t is a single value, at time t?*

In this case, the t subscript refers to the temporally varying variable Srem, for which we note the maximum value. This maximum value (a scalar) is then used to normalize the Srem_t vector, resulting in 1 as highest value.

*P5 L15: Compared instead of inverted?*

We chose to use "evaluated" in this case.

*P6 L2: Consider mentioning that 1.0 is the best possible MEF value for readers not familiar with the metric.*

We agree that this is useful to mention and have changed the manuscript accordingly.

*P9 Eq. 14: What is ET rad?*

We agree that the notation was unclear. We have expanded the equation to show how we determine the relative attenuation due to the SWL-term.

*P9 L29: Remove "was that" P15 L 6: Correct to "led"*

Both cases have been corrected.

*All figures: Avoid analysing results in the figure captions (e.g. models underestimated...), the captions should merely explain what is shown in the figures.*

Done.

*Fig 6: It is hard to see the blue line especially*

This figure is no longer part of the revised manuscript.

[revised manuscript text omitted]